# AVID: ADAPTING VIDEO DIFFUSION MODELS TO WORLD MODELS

## ABSTRACT

Large-scale generative models have achieved remarkable success in a number of domains. However, for sequential decision-making problems, such as robotics, action-labelled data is often scarce and therefore scaling-up foundation models for decision-making remains a challenge. A potential solution lies in leveraging widely-available unlabelled videos to train world models that simulate the consequences of actions. If the world model is accurate, it can be used to optimize decision-making in downstream tasks. Image-to-video diffusion models are already capable of generating highly realistic synthetic videos. However, these models are not action-conditioned, and the most powerful models are closed-source which means they cannot be finetuned. In this work, we propose to adapt pretrained video diffusion models to action-conditioned world models, without access to the parameters of the pretrained model. Our approach, AVID, trains an adapter on a small domain-specific dataset of action-labelled videos. AVID uses a learned mask to modify the intermediate outputs of the pretrained model and generate accurate action-conditioned videos. We evaluate AVID on video game and real-world robotics data, and show that it outperforms existing baselines for diffusion model adaptation. Our results demonstrate that if utilized correctly, pretrained video models have the potential to be powerful tools for embodied AI.

## 1 INTRODUCTION

Large generative models trained on web-scale data have driven rapid improvement in natural language processing (Brown, 2020; Touvron et al., 2023; Achiam et al., 2023), image generation (Rombach et al., 2022), and video generation (OpenAI, 2024). The potential for scaling to unlock progress in sequential-decision making domains, such as robotics, gaming, and virtual agents, has invoked a surge of interest in foundation models for decision-making agents (Reed et al., 2022), particularly for robotics (Brohan et al., 2022; 2023). However, the quantity of action-labelled data in these domains remains a significant bottleneck (Padalkar et al., 2023). This raises the question of to how to utilize widely-available unlabelled videos to bootstrap learning (Baker et al., 2022; Bruce et al., 2024). One promising approach is to use video data to learn a *world model* (Ha & Schmidhuber, 2018), a model the predicts the consequences of actions and acts as a learned simulator. Such a model can be used to optimize decision-making for downstream tasks (Hafner et al., 2021).

Current image and video diffusion models are highly adept at generating text-conditioned synthetic data (Podell et al., 2023; Zhang et al., 2023b; Blattmann et al., 2023). If actions can be expressed as natural language, these models have the potential to be used out-of-the-box for decision-making (Kapelyukh et al., 2023; Zhu et al., 2024b). However, in many real-world domains the core challenge is optimizing low-level actions, such as joint angles in robotics, and therefore using natural language as the only interface is insufficient. To overcome this limitation, one option is to finetune a pre-trained model to condition on low-level actions for a domain-specific dataset (Seo et al., 2022). Another possibility is to apply existing adapter architectures such as ControlNet (Zhang et al., 2023a), to add action-conditioning by modifying the activations inside the original model. However, the parameters for state-of-the-art video diffusion models are usually not available publicly (LumaAI, 2024; OpenAI, 2024; RunwayML, 2024), which rules out these approaches.

In this work we address the problem of exploiting a pretrained video diffusion model to generate action-conditioned predictions *without access to the parameters of the pretrained model*. Inspired

by the recent work from Yang et al. (2024b), we instead assume that we only have access to the noise predictions of the pretrained diffusion model. We propose AVID, a domain-specific adapter that conditions on actions and modifies the noise predictions of the pretrained model to generate accurate action-conditioned predictions. To train the adapter, we assume that we have access to a domain-specific dataset of action-labelled videos. The core contributions of our work are:

- Proposing to adapt pretrained video diffusion models to action-conditioned world models, without access to the parameters of the pretrained model.
- Analyzing the limitations of the adaptation approach proposed by Yang et al. (2024b).
- AVID, a novel approach to adding conditioning to pretrained diffusion models. AVID applies a learned mask to the outputs of of a pretrained model, and combines them with conditional outputs learned by a domain-specific adapter.

We evaluate AVID on video game data, as well as real-world robotics data where we use a 1.4B parameter model trained on internet-scale data as the pretrained model (Xing et al., 2023). Our results show that our approach outperforms existing baselines, and demonstrates that AVID obtains a considerable benefit from using the pretrained model, even with limited access to the model. We advocate for providers of closed-source video models to provide access to intermediate model outputs in their APIs to facilitate the use of adaptation approaches such as AVID.

## 2 PRELIMINARIES

**Denoising Diffusion Probabilistic Models (DDPM)** Diffusion models (Ho et al., 2020; Sohl-Dickstein et al., 2015) are a class of generative models. Consider a sequence of positive noise scales, $0 < \beta_1, \beta_2, \ldots, \beta_N < 1$. In the forward process, for each training data point $\mathbf{x}_0 \sim p_{\text{data}}(\mathbf{x})$, a Markov chain $\mathbf{x}_0, \mathbf{x}_1, \ldots, \mathbf{x}_N$ is constructed such that $p(\mathbf{x}_i \mid \mathbf{x}_{i-1}) = \mathcal{N}(\mathbf{x}_i; \sqrt{1-\beta_i}\mathbf{x}_{i-1}, \beta_i\mathbf{I})$. Therefore, $p_{\alpha_i}(\mathbf{x}_i \mid \mathbf{x}_0) = \mathcal{N}(\mathbf{x}_i; \sqrt{\alpha_i}\mathbf{x}_0, (1-\alpha_i)\mathbf{I})$, where $\alpha_i := \Pi_{j=1}^{i}(1-\beta_j)$. We denote the perturbed data distribution as $p_{\alpha_i}(\mathbf{x}_i) := \int p_{\text{data}}(\mathbf{x})p_{\alpha_i}(\mathbf{x}_i \mid \mathbf{x})\mathrm{d}\mathbf{x}$. The noise scales are chosen such that $\mathbf{x}_N$ is distributed according to $\mathcal{N}(\mathbf{0}, \mathbf{I})$. Define $s(\mathbf{x}_i, i)$ to be the score function of the perturbed data distribution: $s(\mathbf{x}_i, i) := \nabla_{\mathbf{x}_i} \log p_{\alpha_i}(\mathbf{x}_i)$, for all $i$. Samples can be generated from a diffusion model via Langevin dynamics by starting from $\mathbf{x}_N \sim \mathcal{N}(\mathbf{0}, \mathbf{I})$ and following the recursion:

$$\mathbf{x}_{i-1} = \frac{1}{\sqrt{1-\beta_i}}(\mathbf{x}_i + \beta_i s_\theta(\mathbf{x}_i, i)) + \sqrt{\beta_i}\mathbf{z}, \tag{1}$$

where $s_\theta$ is a learned approximation to the true score function $s$, and $\mathbf{z}$ is a sample from the standard normal distribution. If we reparameterize the sampling of the noisy data points according to: $\mathbf{x}_i = \sqrt{\alpha_i}\mathbf{x}_0 + \sqrt{1-\alpha_i}\boldsymbol{\epsilon}$, where $\boldsymbol{\epsilon} \sim \mathcal{N}(\mathbf{0}, \mathbf{I})$, we observe that $\nabla_{\mathbf{x}_i} \log p_{\alpha_i}(\mathbf{x}_i \mid \mathbf{x}_0) = -\frac{\boldsymbol{\epsilon}}{\sqrt{1-\alpha_i}}$. Therefore, we can define the estimated score function in terms of a function $\epsilon_\theta$ that predicts the noise $\boldsymbol{\epsilon}$ used to generate each sample

$$s_\theta(\mathbf{x}_i, i) := -\frac{\epsilon_\theta(\mathbf{x}_i, i)}{\sqrt{1-\alpha_i}}. \tag{2}$$

The noise prediction model $\epsilon_\theta$ is trained to optimize the objective

$$\theta^* = \arg\min_\theta \mathbb{E}_{\mathbf{x}_0 \sim p_{\text{data}}(\mathbf{x}), \boldsymbol{\epsilon} \sim \mathcal{N}(\mathbf{0}, \mathbf{I}), i \sim \mathcal{U}(\{1,2,\ldots,N\})}\left[||\boldsymbol{\epsilon} - \epsilon_\theta(\sqrt{\alpha_i}\mathbf{x}_0 + \sqrt{1-\alpha_i}\boldsymbol{\epsilon}, i)||^2\right]. \tag{3}$$

To add a conditioning signal to the diffusion model, the denoising can be trained with an additional conditioning signal $\epsilon_\theta(\mathbf{x}_i, i, c)$, where $c$ is the desired conditioning signal.

**Probabilistic Adaptation of Diffusion Models** Yang et al. (2024b) proposes to model the problem of diffusion model adaptation via a product of experts. Given a pretrained model $p_{\text{pre}}(\mathbf{x})$ and a small domain specific video model $p_{\text{adapt}}(\mathbf{x})$, the final adapted model is defined as the following product of expert (PoE) distribution:

$$p_{\text{PoE}} := \frac{p_{\text{pre}}(\mathbf{x})p_{\text{adapt}}(\mathbf{x})}{Z},$$

which generates samples that are likely under both of the original models, with the aim of maintaining strong video quality from $p_{\text{pre}}(\mathbf{x})$ and the desired domain-specific quality from $p_{\text{adapt}}(\mathbf{x})$. In practice, $p_{\text{PoE}}$ is intractable. To sample from the PoE, the authors propose to compose their scores:

$$\epsilon_{\text{PoE}}(\mathbf{x}_i, i, c) := \epsilon_{\text{pre}}(\mathbf{x}_i, i, c) + \epsilon_{\text{adapt}}(\mathbf{x}_i, i, c),$$

and pass the combined score to a DDPM sampler.

## 3 ADAPTING VIDEO DIFFUSION MODELS TO WORLD MODELS (AVID)

### 3.1 PROBLEM SETTING

In the context of video diffusion models each datapoint is a video, $\mathbf{x} = [x^0, x^1, \ldots, x^{T-1}]$, where $x^\tau$ indicates a video frame. Note that we use superscript indices, $x^\tau$, to indicate steps in time, and subscript indices, $\mathbf{x}_i$, to indicate steps in a diffusion process. We assume that we have access to a pretrained image-to-video diffusion model $\epsilon_{\text{pre}}$, that is trained on web-scale data. Given an initial image, $x^0$, the image-to-video model $\epsilon_{\text{pre}}$ generates a synthetic video $\hat{\mathbf{x}} = [x^0, \hat{x}^1, \ldots, \hat{x}^{T-1}]$.

We consider each video to be a sequence of observations generated by a partially observable Markov decision process (POMDP) (Kaelbling et al., 1998), with corresponding action sequence $\mathbf{a} = [a^0, a^1, \ldots, a^{T-1}]$. For each domain, we assume access to a dataset of action-labelled videos, $\mathcal{D} = \{(\mathbf{x}, \mathbf{a}), \ldots\}$. Given a new initial image, $x^0$, and action sequence $\mathbf{a}$, the goal is to generate a synthetic video $\hat{\mathbf{x}}$ that accurately depicts the ground-truth realization of the actions.

To utilize the pretrained model $\epsilon_{\text{pre}}$, we need to add action-conditioning to the model as accurate videos cannot be generated without the actions. Adding a new conditioning signal to a pretrained diffusion model is well-explored in previous works (Zhang et al., 2023a; Mou et al., 2024; Mokady et al., 2023). However, most existing works assume access to the parameters of the pretrained model. In this work, we assume that we *do not have access to the parameters of the pretrained model*.

### 3.2 LIMITATIONS OF NAIVE ADAPTATION OF YANG ET AL. (2024B)

Yang et al. (2024b) propose to adapt a pretrained diffusion model to a specific use-case without access it its weights by composing (omitting the prior strength $\lambda$):

$$\epsilon_{\text{PoE}}(\mathbf{x}_i, i, c) := \epsilon_{\text{pre}}(\mathbf{x}_i, i, c) + \epsilon_{\text{adapt}}(\mathbf{x}_i, i, c) \tag{4}$$

However, this method has a fundamental limitation that it will produce biased samples. To see this, consider the following forward diffusion process

$$d\mathbf{x} = -\frac{1}{2}\beta(t)\mathbf{x}dt + \sqrt{\beta(t)}dW,$$

where $t$ indicates continuous time. Assume the intial target distribution at $t = 0$ is $p_0(\mathbf{x}) := p_{\text{PoE}} = \frac{p_{\text{pre}}(\mathbf{x})p_{\text{adapt}}(\mathbf{x})}{Z}$. Notice that the transition kernel can be written as (Särkkä & Solin, 2019):

$$p(\mathbf{x}_t|\mathbf{x}_0) = \mathcal{N}(\mathbf{x}_t; \mathbf{x}_0 e^{-\frac{1}{2}\int_0^t \beta(s)ds}, I - Ie^{-\int_0^t \beta(s)ds})$$

Therefore, for $\forall t > 0$, the resulting distribution is the convolution $p_t(\mathbf{x}_t) = p_0(\mathbf{x}_0) * \mathcal{N}(\mathbf{x}_t - \mathbf{x}_0 e^{-\frac{1}{2}\int_0^t \beta(s)ds}; 0, I - Ie^{-\int_0^t \beta(s)ds})$. Due to the fact that convolution does not distribute over multiplication, the resulting score $s(\mathbf{x}_t, t)$ cannot be expressed as a sum of the two individual score functions:

$$s(\mathbf{x}_t, t) = \nabla_{\mathbf{x}_t} \log \left[ p_0(\mathbf{x}_0) * \mathcal{N}(\mathbf{x}_t - \mathbf{x}_0 e^{-\frac{1}{2}\int_0^t \beta(s)ds}; 0, I - Ie^{-\int_0^t \beta(s)ds}) \right]$$

$$\neq \nabla_{\mathbf{x}_t} \log \left[ p_{0,\text{pre}}(\mathbf{x}_0) * \mathcal{N}(\mathbf{x}_t - \mathbf{x}_0 e^{-\frac{1}{2}\int_0^t \beta(s)ds}; 0, I - Ie^{-\int_0^t \beta(s)ds}) \right]$$

$$+ \nabla_{\mathbf{x}_t} \log \left[ p_{0,\text{adapt}}(\mathbf{x}_0) * \mathcal{N}(\mathbf{x}_t - \mathbf{x}_0 e^{-\frac{1}{2}\int_0^t \beta(s)ds}; 0, I - Ie^{-\int_0^t \beta(s)ds}) \right]$$

As a result, the true noise prediction of the target distribution also cannot be expressed as a sum: $\epsilon_{\text{PoE}}(\mathbf{x}_t, t, c) \neq \epsilon_{\text{pre}}(\mathbf{x}_t, t, c) + \epsilon_{\text{adapt}}(\mathbf{x}_t, t, c)$. Therefore, the composition in Equation (4) does not hold and will result in biased samples Du et al. (2023). In the following section, we propose AVID to overcome this limitation. Rather than attempting to compose two independently trained models, AVID uses the outputs of the pretrained model to train an adapter that directly optimizes the denoising loss.

### 3.3 ADAPTING VIDEO DIFFUSION MODELS TO WORLD MODELS (AVID)

AVID is a new approach for diffusion model adaptation that does not require access to the pretrained model. The motivation for AVID is that while pretrained image-to-video models can generate *realistic* videos, they cannot generate videos that are *accurate* with respect to a given sequence of actions.

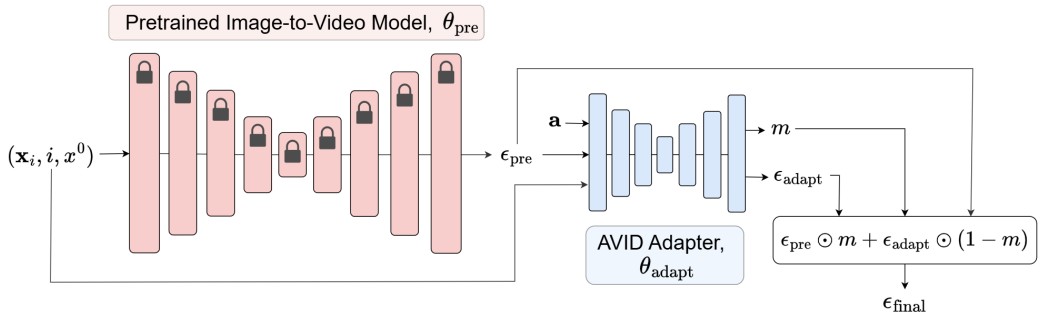

Figure 1: Overview of AVID world model adapter architecture.

To achieve accuracy, the pretrained model must be guided towards the correct generation for the action sequence. However, as our experiments show, techniques such as classifier(-free) guidance do not perform well in this setting. AVID addresses this by training a lightweight adapter to adjust the output of the pretrained model to achieve accurate action-conditioned video predictions.

AVID trains an adapter that takes the output of the pretrained model as an input. AVID learns to generate a mask, and uses this mask to combine the outputs of the pretrained model with those of the adapter. The final combined output is used to compute the standard denoising loss, and the adapter parameters are trained to optimize this loss. We train the AVID adapter using samples $(\mathbf{x}, \mathbf{a})$ from the action-labelled dataset. For the pretrained image-to-video model $\epsilon_{\text{pre}}$ we assume that we do not have to access to it's parameters, $\theta_{\text{pre}}$, but we can run inference using the model to obtain it's noise predictions. To do this, we input a noisy video, $\mathbf{x}_i \in \mathbb{R}^{T \times h \times w \times c}$, initial image, $x^0 \in \mathbb{R}^{h \times w \times c}$, and diffusion step, $i$, to the pretrained image-to-video model to obtain its noise prediction, $\epsilon_{\text{pre}}(\mathbf{x}_i, i, x_0) \in \mathbb{R}^{T \times h \times w \times c}$. The parameters of the pretrained model $\epsilon_{\text{pre}}$ are not modified.

The adapter that we train is a 3D UNet (Ho et al., 2022c; Ronneberger et al., 2015) consisting of a sequence of spatio-temporal blocks with residual connections. Each spatio-temporal block consists of a 3D convolution, spatial attention, and temporal attention (Ho et al., 2022c). The UNet takes as input a tensor of shape $\mathbb{R}^{T \times h \times w \times 3c}$. The input consists of the noisy video, $\mathbf{x}_i$, the output of the pretrained model, $\epsilon_{\text{pre}}(\mathbf{x}_i, i, x_0)$, and the initial image $x^0$ repeated $T$ times across the time dimension. These three inputs are concatenated channel-wise to create the input tensor.

The adapter is also conditioned on the diffusion step $i$ and the sequence of actions $\mathbf{a}$. For noisy video $\mathbf{x}_i$ we embed the diffusion timestep according to a learned embedding table to get the diffusion step embedding $e_i$. For each timestep $\tau$ of the noisy video $\mathbf{x}_i$ we embed the corresponding action $a^\tau$ to compute the action embedding $e_a^\tau$ using an embedding table for discrete actions or a linear layer for continuous actions. In each block, these two embeddings are concatenated and processed by an MLP to compute the scale and shift parameters, $\gamma^\tau$ and $\beta^\tau$, for the $\tau^{th}$ frame. These parameters scale and shift the feature maps of the $\tau^{th}$ frame after each 3D convolution (Perez et al., 2018).

To inject the correct action-conditioning into the predictions made by the pretrained model, the adapter needs to erase incorrect motions predicted by the pretrained model and add the correct motion. To facilitate this, the adapter outputs a tensor of shape $\mathbb{R}^{T \times h \times w \times (c+1)}$ consisting of a mask, $m \in \mathbb{R}^{T \times h \times w \times 1}$ which is bounded between 0 and 1 by a sigmoid layer, and the noise prediction from the adapter $\epsilon_{\text{adapt}}$. The mask is then used to combine the noise predictions from the pretrained model and the adapter according to:

$$\epsilon_{\text{final}}(\mathbf{x}_i, \mathbf{a}, i, x_0) = \epsilon_{\text{pre}} \odot m + \epsilon_{\text{adapt}} \odot (1 - m), \tag{5}$$

where $\odot$ denotes the Hadamard product, and the mask is broadcast across all $c$ channels. The parameters of the adapter model, $\theta_{\text{adapt}}$, are optimized to minimise the standard unweighted denoising loss using $\epsilon_{\text{final}}$:

$$\mathcal{L}(\theta_{\text{adapt}}) = \mathbb{E}_{(\mathbf{x}, \mathbf{a}) \sim \mathcal{D}, \epsilon \sim \mathcal{N}(\mathbf{0}, \mathbf{I}), i \sim \mathcal{U}(\{1, 2, ..., N\})} \| \epsilon_{\text{final}}(\mathbf{x}_i, \mathbf{a}, i, x_0) - \epsilon \|^2. \tag{6}$$

In the case where $\epsilon_{\text{pre}}$ is a latent diffusion model, we assume that we can also run inference on the corresponding encoder and decoder. For each training example, we first encode the video: $\mathbf{z} = \text{enc}(\mathbf{x})$ where $\mathbf{z} \in \mathbb{R}^{T \times h' \times w' \times c'}$, and add noise to this latent representation of the video to produce

noisy latent $\mathbf{z}_i$. The rest of the pipeline proceeds in the same manner, except that the initial image $x^0$ is replaced by the latent corresponding to the first frame, $z^0$, and the noisy video $\mathbf{x}_i$ is replaced by the noisy latent $\mathbf{z}_i$. The adapter loss in Equation 6 predicts the noise added to the latent sample, and we decode the sampled latent to generate the final video.

# 4 EXPERIMENTS

We evaluate AVID in two different domains with different pretrained base models. Details about the datasets are provided in Appendix B.1 and details about the pretrained models are in Appendix B.2. As the focus of our work is on training lightweight adapters with limited compute, we compare adapter models with limited parameters under a computational budget.

**Procgen** The pretrained model is a pixel-space image-to-video diffusion model (Ho et al., 2022c) trained on videos sampled from 15 out of the 16 procedurally generated games of Procgen (Cobbe et al., 2020), excluding the 16th game Coinrun. The adaptation approaches are trained using an action-labelled dataset sampled from Coinrun. At each timestep, the discrete action is one of 15 possible keypad inputs, and the models are trained on sequences of 10 frames. Each adaptation approach is limited to 3 days of training on a single A100 GPU. We test three dataset sizes, *Coinrun100k/500k/2.5M*, evaluate the models on a held-out test set of initial frames and actions.

**RT1 + DynamiCrafter** In this benchmark, the pretrained model is DynamiCrafter (Xing et al., 2023) which is currently one of the best performing image-to-video models in the VBench image-to-video leaderboard (Huang et al., 2024). DynamiCrafter is a latent image-to-video diffusion model that uses the autoencoder from Stable Diffusion (Rombach et al., 2022) and a 1.4B parameter 3D UNet trained on web-scale video data. As DynamiCrafter is a latent diffusion model, we assume that we can run inference on the encoder and decoder as described in Section 3.3. For the action-labelled dataset we use the RT1 dataset (Brohan et al., 2022) which consists of real-world robotics videos. The action at each step is a 7 dimensional continuous vector corresponding to the movement and rotation of the end effector and opening or closing of the gripper. The models are trained on trajectories of 16 frames. Each adaptation approach is limited to 7 days of training on $4\times$ A100 GPUs, and is evaluated using a held-out test set of ground truth trajectories.

## 4.1 BASELINES

We compare AVID with several baselines, both with and without access to the parameters of the pretrained model. Further details, including hyperparameter tuning, are in Appendix B.5.

**Full access to pretrained model parameters:**

- *Action-Conditioned Finetuning* – tunes all of the parameters of the pretrained model on the action-conditioned dataset. To add action-conditioning, we first compute an action embedding according to Section 3.3. For Procgen and RT1, we concatenate and add the action embeddings with the time step embeddings respectively.

- *Language-Conditioned Finetuning* – finetunes the pretrained model using a language description of each video. The language is embedded using CLIP (Radford et al., 2021) and conditioned upon using cross-attention following Xing et al. (2023).

- *ControlNet* (Zhang et al., 2023a) – freezes the parameters of the pretrained model and makes a *trainable copy* of its UNet encoder. The trainable part of the model is conditioned on a new signal, and connected to the decoder in the original model via convolutions which are initialised to zero. In our work, we use ControlNet to add action-conditioning.

- *ControlNet Small* – ControlNet still has a large number of trainable parameters. ControlNet Small freezes the pretrained model in the same way as ControlNet, but reduces the number of trainable parameters to a similar amount to AVID by decreasing the number of channels in the layers of the UNet encoder. A learned projection at each layer projects the activations of the smaller model to match the number of channels in the pretrained base model.

**No access to pretrained model parameters** For a fair comparison to AVID, we evaluate the following approaches which either do not leverage the pretrained model or, like AVID, assume access to only noise predictions from the pretrained model:

- *Action-Conditioned Diffusion* – We train an action-conditioned diffusion model $\epsilon_\theta(\mathbf{x}_i, \mathbf{a}, i, x_0)$ from scratch, with the same number of parameters and same UNet architecture as AVID.

- *Classifier Guidance* (Dhariwal & Nichol, 2021) – We train a classifier $f_\phi(\mathbf{a}|\mathbf{x}_i)$ on noisy images $\mathbf{x}_i$ to predict actions. With weighting $w$, the classifier is used to steer the diffusion sampling process towards samples consistent with the actions. The final noise prediction becomes:

$$\bar{\epsilon}_{\text{final}}(\mathbf{x}_i, \mathbf{a}, i, x_0) = \epsilon_{\text{pre}}(\mathbf{x}_i, i, x_0) - \sqrt{1 - \bar{\alpha}_t}\, w \nabla_{\mathbf{x}_i} \log f_\phi(\mathbf{a}|\mathbf{x}_i).$$

- *Product of Experts* – Inspired by Yang et al. (2024b), we add action conditioning to a pretrained video diffusion model by adding its score to an action-conditioned model. We train a small video diffusion model on action conditioned data $\epsilon_{\text{adapt}}(\mathbf{x}_i, \mathbf{a}, i, x_0)$, and compute the final denoising prediction as a weighted sum of predictions from the pretrained and action-conditioned models:

$$\bar{\epsilon}_{\text{final}}(\mathbf{x}_i, \mathbf{a}, i, x_0) = \lambda_p \epsilon_{\text{adapt}}(\mathbf{x}_i, \mathbf{a}, i, x_0) + (1 - \lambda_p)\epsilon_{\text{pre}}(\mathbf{x}_i, i, x_0),$$

where $\lambda_p$ controls the strength of the pretrained prior during video generation.

- *Action Classifier-Free Guidance* (Ho & Salimans, 2021) – We train a small action-conditioned diffusion model $\epsilon_{\text{adapt}}(\mathbf{x}_i, \mathbf{a}, i, x_0)$ while randomly removing the action conditioning ($\mathbf{a} = \varnothing$) during training with probability $p = 0.2$. We then compute the final denoising prediction as:

$$\bar{\epsilon}_{\text{final}}(\mathbf{x}_i, \mathbf{a}, i, x_0) = \epsilon_{\text{pre}}(\mathbf{x}_i, i, x_0) + \lambda_a \left( \epsilon_{\text{adapt}}(\mathbf{x}_i, \mathbf{a}, i, x_0) - \epsilon_{\text{adapt}}(\mathbf{x}_i, \mathbf{a} = \varnothing, i, x_0) \right).$$

Note that unlike standard classifier-free guidance this approach combines predictions from two separate models, $\epsilon_{\text{pre}}$ and $\epsilon_{\text{adapt}}$, which are trained on different data.

## 4.2 EVALUATION

For evaluation, we use a set of 1024 held-out test videos and their corresponding action sequences. We generate 1024 synthetic videos by conditioning the models on each initial frame and action sequence, and compare the generated videos against the ground truth using the following metrics:

- *Action Error Ratio* – To assess the consistency between the videos and the action sequences, we train a model to predict actions from real videos. The Action Error Ratio is the ratio of errors obtained by using this model on generated videos, divided by the error obtained on real videos. More details are in Appendix B.3.

- *FVD* (Cobbe et al., 2019) – measures the similarity between feature distributions of generated and real video sequences in the I3D network (Carreira & Zisserman, 2017), accounting for both spatial quality and temporal coherence.

- *FID* (Heusel et al., 2017) – compares the feature distributions of all real and generated images in the Inception-v3 network (Szegedy et al., 2015), measuring the quality and variety of the images.

- *SSIM* (Wang et al., 2004) – compares each ground truth and generated image, focusing on luminance, contrast, and structural information.

- *PSNR* (Hore & Ziou, 2010) – is computed using the mean squared error between each ground truth and generated video, and thus computes the distance between the videos in pixel-space.

- *LPIPS* (Zhang et al., 2018) – compares the similarity between the features representations of each ground truth and predicted image using the VGG network (Simonyan & Zisserman, 2015).

We bold results within 2% of the best performance in each model size category. We also compute normalized metrics by normalizing each value between 0 and 1. Details are in Appendix B.4.

## 4.3 RESULTS

**Qualitative Results** Examples of generated videos are in Figures 2 and 3, with further examples in Appendix A.5. We observe for both RT1 and Coinrun that the action-conditioned diffusion model trained from scratch fails to maintain consistency with the original conditioning image: objects in the initial image disappear in later frames in the video. The videos generated by PoE are blurry, and sometimes appear like two superimposed videos. In contrast, the videos generated by AVID are consistent throughout. In both AVID and action-conditioned diffusion, we observe that the motion in the generated videos is accurate compared to the ground truth motion. The pretrained base models do not generate accurate videos for either domain (Appendix A.5). The masks generated by AVID

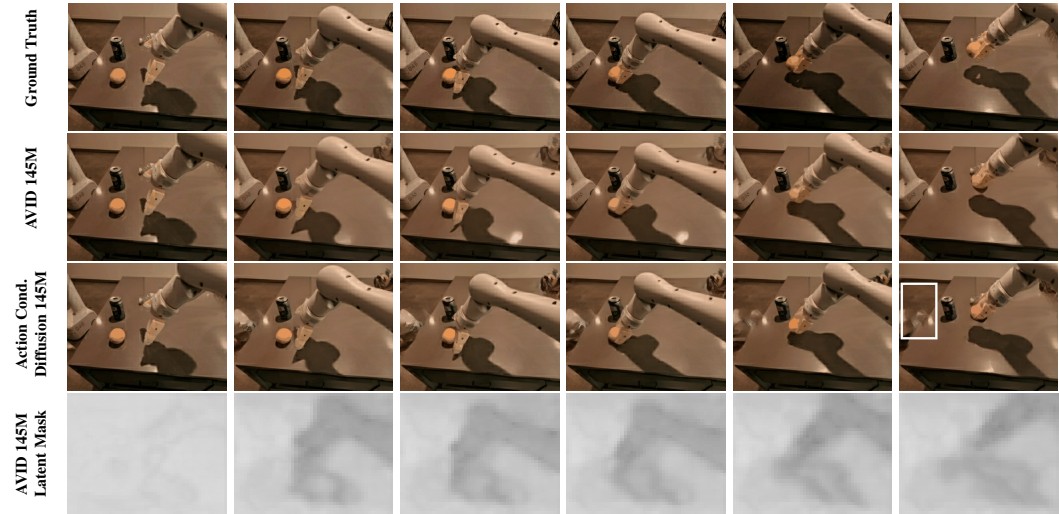

Figure 2: Top three rows: Examples of videos generated for RT1 (extended in Figure 7, Appendix A.5). Bottom row: Mask generated in downsampled latent space by AVID. White indicates the mask is set to 1 and black indicates the mask is set to 0.

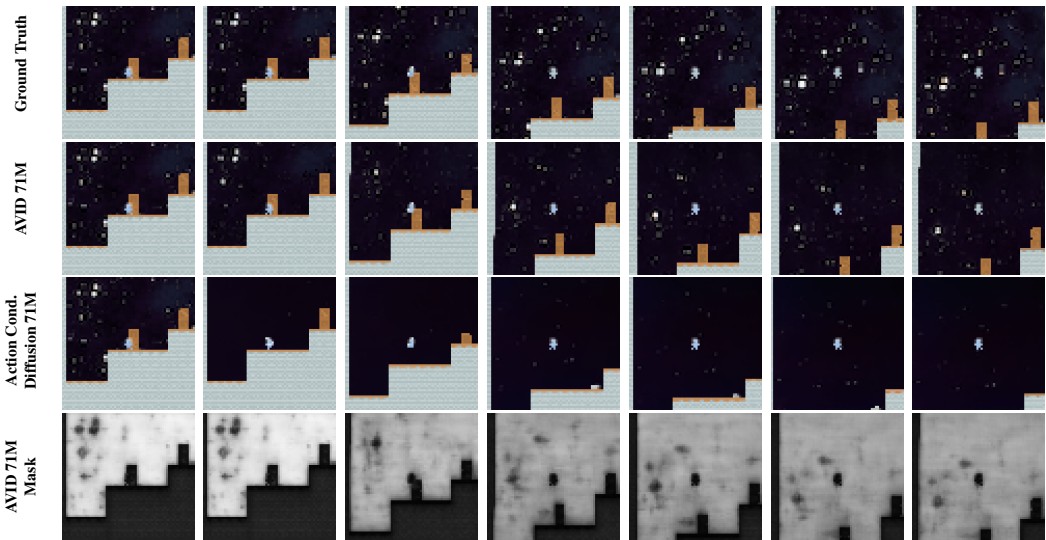

Figure 3: Top three rows: Examples of videos generated for Coinrun 500k (extended in Figure 8, Appendix A.5). Bottom row: Mask generated by AVID where white indicates the mask is set to 1 and black indicates the mask is set to 0.

are visualised in Figures 2 and 3. The lighter parts of the mask show that the pretrained models predictions are predominantly used by AVID for maintaining background textures. The mask is reduced nearer to 0 around the robot arm in RT1 and the character in Coinrun, showing that the adapter outputs are predominantly used for action-relevant parts of the video. In Appendix A.6 we show that AVID can be used to generate predictions for different actions given the same initial frame.

**Quantitative Results** To evaluate overall performance, in Figures 4a and 4b we plot the normalized performance averaged across all evaluation metrics. AVID obtains similar or slightly better overall performance compared to ControlNet/ControlNet Small on both Coinrun500k and RT1. Note that unlike ControlNet variants, AVID does not require access to the weights of the original pretrained model. For RT1, we observe that training an action-conditioned diffusion model performs slightly worse than AVID at the largest model size. For Coinrun500k, AVID significantly outperforms action-conditioned diffusion at the larger model size. As the number of trainable parameters

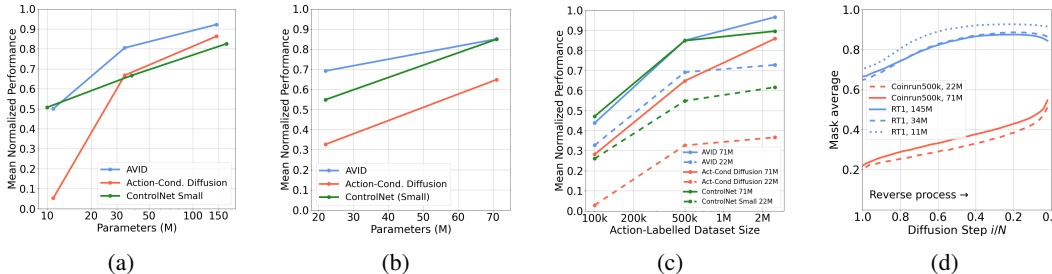

(a)  (b)  (c)  (d)

Figure 4: (a) RT1 averaged normalized performance versus parameter count. (b) Coinrun500k averaged normalized performance versus parameter count. (c) Coinrun averaged normalized performance versus dataset size. Details on metric normalization are in Appendix B.4. (d) Average mask ($m$) values of AVID throughout diffusion process.

| | Method | Action Err. Ratio ↓ | FVD ↓ | FID ↓ | SSIM ↑ | LPIPS ↓ | PSNR ↑ |
|---|---|---|---|---|---|---|---|
| **Medium** | AVID (Ours) (22M) | **1.257** | **28.5** | **8.07** | **0.666** | **0.192** | **22.4** |
| | Action-Conditioned Diffusion (22M) | 1.500 | 41.6 | 8.95 | 0.562 | 0.284 | 18.7 |
| | Product of Experts (22M) | 1.601 | 101.8 | 9.96 | 0.584 | 0.281 | 19.1 |
| | Action Classifier-Free Guidance (22M) | 1.966 | 191.1 | 13.09 | 0.454 | 0.3495 | 17.5 |
| | ControlNet-Small (22M) | 1.465 | 34.7 | 8.49 | 0.652 | 0.205 | 21.9 |
| **Large** | AVID (Ours) (71M) | **1.154** | 23.1 | 7.33 | 0.713 | 0.161 | 23.8 |
| | Action-Conditioned Diffusion (71M) | 1.216 | 31.3 | 7.78 | 0.648 | 0.220 | 20.9 |
| | Product of Experts (71M) | 1.247 | 84.8 | 9.07 | 0.651 | 0.229 | 21.0 |
| | Action Classifier-Free Guidance (71M) | 2.079 | 188.7 | 13.05 | 0.463 | 0.341 | 17.7 |
| | ControlNet (71M) | 1.418 | **18.5** | **7.16** | **0.758** | **0.128** | **25.5** |
| **Full** | Pretrained Base Model (97M) | 3.855 | 204.0 | 13.03 | 0.451 | 0.352 | 17.3 |
| | Classifier Guidance | 4.070 | 192.9 | 12.68 | 0.450 | 0.351 | 17.4 |
| | Action-Conditioned Finetuning (97M) | **1.227** | **14.1** | **6.52** | **0.761** | **0.120** | **25.8** |

Table 1: Results for Coinrun 500k dataset. Methods trained for 3 days on a single A100. Shading indicates method requires access to the model parameters. Brackets indicate trainable parameters.

| | Method | Action Err. Ratio ↓ | FVD ↓ | FID ↓ | SSIM ↑ | LPIPS ↓ | PSNR ↑ |
|---|---|---|---|---|---|---|---|
| **Small** | AVID (Ours) (11M) | 2.572 | 54.0 | 4.344 | **0.811** | **0.166** | **24.5** |
| | Action-Conditioned Diffusion (11M) | **2.238** | 80.4 | 5.329 | 0.767 | 0.226 | 22.9 |
| | Product of Experts (11M) | 2.859 | 89.9 | 5.276 | 0.790 | 0.201 | 23.6 |
| | Action Classifier-Free Guidance (11M) | 3.503 | 104.8 | 4.858 | 0.737 | 0.217 | 22.2 |
| | ControlNet-Small (10M) | 2.640 | **38.6** | **3.730** | 0.811 | 0.169 | 23.4 |
| **Medium** | AVID (Ours) (34M) | 1.907 | 38.7 | 3.645 | **0.831** | **0.150** | **25.0** |
| | Action-Conditioned Diffusion (34M) | **1.737** | 36.7 | 4.038 | 0.813 | 0.172 | 24.3 |
| | Product of Experts (34M) | 2.421 | 61.7 | 4.533 | 0.813 | 0.175 | 24.4 |
| | Action Classifier-Free Guidance (34M) | 3.129 | 71.4 | 4.690 | 0.748 | 0.205 | 22.6 |
| | ControlNet-Small (38M) | 2.227 | **35.0** | **3.539** | **0.821** | 0.162 | 24.0 |
| **Large** | AVID (Ours) (145M) | 1.609 | 39.3 | 3.436 | **0.842** | **0.142** | **25.3** |
| | Action-Conditioned Diffusion (145M) | **1.384** | **24.9** | 3.504 | 0.817 | 0.153 | 24.6 |
| | Product of Experts (145M) | 1.947 | 47.0 | 4.026 | 0.819 | 0.160 | 24.8 |
| | Action Classifier-Free Guidance (145M) | 3.188 | 79.3 | 4.775 | 0.748 | 0.205 | 22.8 |
| | ControlNet-Small (170M) | 1.779 | 30.0 | **3.375** | **0.832** | 0.153 | 24.4 |
| **Full** | Pretrained Base Model (1.4B) | 4.183 | 237.6 | 5.432 | 0.712 | 0.228 | 20.6 |
| | Classifier Guidance | 4.182 | 213.1 | 6.005 | 0.683 | 0.250 | 19.8 |
| | ControlNet (654M) | 1.708 | 27.1 | 3.248 | 0.836 | 0.148 | 24.5 |
| | Action-Conditioned Finetuning (1.4B) | **1.297** | **24.2** | **2.965** | **0.852** | **0.134** | **25.6** |
| | Language-Conditioned Finetuning (1.4B) | 3.859 | 33.7 | 3.511 | 0.812 | 0.177 | 22.1 |

Table 2: Quantitative results for RT1 dataset. Methods trained for 7 days on $4\times$ A100. Shading indicates method requires access to the model parameters. Brackets indicate trainable parameters.

is reduced, the performance of action-conditioned diffusion declines much more quickly than AVID in both domains, and therefore AVID is considerably stronger at smaller model sizes. In Figure 4c we plot the overall performance of these three approaches against the Coinrun dataset size. A similar trend is observed for all approaches as dataset size is modified.

In Figure 4d we plot the average mask value at each step of the diffusion process. On RT1 AVID has a higher mask value, and therefore uses the pretrained model more heavily than on Coinrun. This is likely because the backgrounds in RT1 are mostly static and DynamiCrafter is a strong model. We see that the mask values are lower at diffusion steps where the noise level is high. This indicates that the adapter model is more responsible for generating low-frequency information, such as the positions of objects, which is defined early in the reverse process. Towards the end of the reverse process, where fine details are generated (Ho et al., 2020), the pretrained model is relied on more.

The values for every evaluation metric are reported in Tables 1 and 2. For Coinrun500k, AVID performs the best for every evaluation metric at the smaller 22M model size. For the larger model size of 71M, AVID performs the second best for most metrics to ControlNet, but obtains the best performance for Action Error Ratio. In RT1, AVID is the strongest in the metrics that make frame-wise comparisons (SSIM/LPIPS/PSNR) across all model sizes. In our setting, where the goal is to generate accurate videos according to the input action sequence, these metrics are more suitable than comparing the overall distribution of images and videos (i.e. FID and FVD) (Zhu et al., 2024a). ControlNet Small generally obtains the best performance for FVD and FID, while Action-Conditioned Diffusion is consistently the best for Action Error Ratio. Poor performance was obtained for Classifier Guidance and Action Classifier-Free Guidance across both domains. For standard implementations of classifier(-free) guidance, the unconditional model is trained on the trained on data from the target domain. In contrast, in our setting the pretrained models are not trained on data from Coinrun or RT1 which may explain the lackluster performance of these methods. Product of Experts also performed poorly (PoE). However, for PSNR and SSIM, PoE did slightly outperform both of the models from which it is composed.

Across both domains, finetuning the pretrained model with action conditioning is the strongest baseline. However, like the ControlNet variants, this requires access to the weights of the pretrained model which we assume we do not have access to. In comparison, finetuning with language instead of action conditioning results in poor performance in all metrics except FID and FVD, demonstrating that fine-grained action conditioning is necessary to generate accurate synthetic videos.

**AVID Ablations**  We evaluate the following two ablations of AVID: *No Mask* (NM): The output adapter does not output a mask. Instead, the outputs of the pretrained model and adapter are directly added: $\epsilon_{\text{final}} = \epsilon_{\text{pre}} + \epsilon_{\text{adapt}}$. *No Conditioning* (NC): The adapter is no longer conditioned on the output of the pretrained model, $\epsilon_{\text{pre}}$. Results for the ablations are in Table 3. For NM, performance across most metrics is worse for RT1, but similar for Coinrun500k. For NC, the performance on most metrics gets significantly worse for both Coinrun500k and RT1. Output conditioning enables AVID to observe errors in the pretrained output and then immediately make corrections. In contrast, NC has slower feedback to correct the pretrained model output as discussed in Zavadski et al. (2023).

| | Method | Action Err. Ratio ↓ | FVD ↓ | FID ↓ | SSIM ↑ | LPIPS ↓ | PSNR ↑ |
|---|---|---|---|---|---|---|---|
| Coinrun 500k Large | AVID (Ours) (71M) | **1.154** | 23.1 | 7.33 | **0.713** | **0.161** | **23.8** |
| | No Mask (71M) | **1.136** | **22.2** | **7.15** | **0.721** | **0.158** | **24.0** |
| | No Conditioning (71M) | **1.141** | 27.8 | 8.02 | 0.682 | 0.189 | 22.3 |
| RT1 Large | AVID (Ours) (145M) | **1.609** | 39.3 | **3.436** | **0.842** | **0.142** | **25.3** |
| | No Mask (145M) | 1.769 | 44.1 | 3.533 | **0.836** | **0.146** | **25.3** |
| | No Conditioning (145M) | 1.775 | **36.2** | 3.550 | 0.827 | 0.149 | 25.0 |

Table 3: Results for AVID ablations (extended in Table 6, Appendix A.2).

## 5 RELATED WORK

**Diffusion for Decision-Making**  Many works utilise diffusion models for generating actions (Pearce et al., 2023; Janner et al., 2022). However, the focus of our work is on generating action-conditioned synthetic data for world modelling (Ha & Schmidhuber, 2018), which can be used downstream for planning (Hafner et al., 2021). SynTHER (Lu et al., 2024) employs an unconditional diffusion model to generate synthetic data for reinforcement learning (RL). Other works (Alonso et al., 2024; Yang et al., 2024a; Rigter et al., 2024; Hu et al., 2023; Zhang et al., 2024) utilise diffusion models to train action-conditioned world models. All of these works train the diffusion model from scratch. In our work, the focus is on making effective use of a pretrained video diffusion model to leverage web-scale pretraining. Most related is Pandora (Xiang et al., 2024) which integrates an LLM and text-to-video diffusion model to generate videos conditioned on actions described as natural language. While natural language is suitable for describing high-level actions, it is inadequate for our goal of modelling low-level actions (McCarthy et al., 2024).

**Video Pre-Training for World Models**  To address the scarcity of action-labeled real-world data, several studies have investigated using unlabeled videos to enhance the efficiency of world model training. Seo et al. (2022) and Wu et al. (2024) pre-train an autoregressive video prediction model, which is later fine-tuned with action-labeled data. Mendonca et al. (2023) develop a world model from videos of humans by defining a high-level action space that is shared between both embodiments. Another approach involves learning latent actions from videos (Bruce et al., 2024; Schmidt

& Jiang, 2024). Our work integrates video pre-training by utilizing a pre-trained video diffusion model, with the distinction that we do not have access to the weights of the original model.

**Adding Controllability to Diffusion Models**  Classifier guidance (Dhariwal & Nichol, 2021) adds conditioning to a pretrained diffusion model using a separate classifier. Classifier-free guidance (Ho & Salimans, 2021) achieves stronger empirical performance but requires the model to be trained with conditioning signals, rendering it unsuitable for post-hoc application to a pretrained model.

Recent advancements include ControlNet (Zhang et al., 2023a) and T2I-Adapter (Mou et al., 2024), which introduce conditional control to pretrained diffusion models by freezing the original UNet and injecting additive signals into the pretrained network. ControlNet-XS (Zavadski et al., 2023) extends this concept by increasing the interactions between the pretrained and adapter networks. Li et al. (2023) incorporates additional trainable layers into the frozen pretrained UNet. These techniques have been applied to introduce controls such as language (Xing et al., 2024), optical flow (Hu & Xu, 2023), and depth maps (Chen et al., 2023b) into video models. However, they are not applicable in our context as they require access to the original model's weights, which we assume are inaccessible. Textual inversion (Gal et al., 2023) and null text inversion (Mokady et al., 2023) optimize text embeddings to add controllability to text-conditioned diffusion models. However, this requires backpropagation through the pretrained model which is not feasible in our setting. Cascaded diffusion models (Ho et al., 2022a;b), train a sequence of diffusion models, which each condition on the outputs of the previous diffusion model. However, the focus of these works is improving the spatial and temporal resolution at each step, whereas our focus is on incorporating a new conditioning signal, actions, that was absent in the pretrained model.

**Compositional Generative Models**  Our work is related to compositional generative models, where generative models are combined probabilistically (Liu et al., 2022; Nie et al., 2021; Du et al., 2020). RoboDreamer applies this idea to world modeling by decomposing language conditioning into multiple components to generate several predictions that are then combined (Zhou et al., 2024). Yang et al. (2024b) modifies the style of a pretrained video diffusion model by combining it's output with a domain-specific diffusion model that is trained independently. Unlike Yang et al. (2024b), the focus of our work is on adding action-conditioning to a pretrained model, and our experiments demonstrate that the approach proposed by Yang et al. (2024b) works poorly in our setting.

## 6  DISCUSSION

**Limitations**  Adding the AVID adapter increases the inference compute required compared to the pretrained model. AVID adapters are tailored to a specific pretrained model and therefore cannot be composed with different models. Developing a method that works across different pretrained models is an exciting direction for future work. AVID does not require access to pretrained model weights, but it does require access to intermediate predictions during denoising, including the outputs of the encoder and decoder in the case of latent diffusion. Many closed-source APIs do not provide access to these quantities, and therefore we advocate for model providers to provide API access to the outputs of the denoising model and autoencoder to facilitate more flexible use of their models.

In the RT1 domain we found that training an action-conditioned diffusion model from scratch resulted in the best Action Error Ratio, despite the videos being less visually accurate. For some downstream applications, action consistency might be the most important performance metric. If this is the case, training from scratch may be the preferred approach for some domains.

**Conclusion**  We introduced the novel problem of adapting pretrained video diffusion models to action-conditioned world models without requiring access to the pretrained model's parameters. Our proposed approach, AVID, generates accurate videos with similar performance to ControlNet variants without requiring access to the pretrained model parameters. AVID obtains superior overall performance to existing baselines that do not require access to the internals of the pretrained model. Our results demonstrate that AVID benefits from the pretrained model by maintaining better consistency with the initial image across both pixel-space and latent diffusion models.

As general image-to-video diffusion models continue to advance in capability, our findings highlight the considerable potential of adapting these models to world models that are suitable for planning and decision-making. This work represents an initial step in that direction. In future research, we aim to use synthetic data generated by AVID adapters for planning tasks. We also wish to explore using AVID adapters to add new conditioning signals to pretrained models other than actions.

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

# A ADDITIONAL RESULTS

## A.1 COINRUN100K AND COINRUN 2.5M RESULTS

Tables 4 and 5 contain results for Coinrun datasets of different sizes (100k and 2.5M). The results in the main paper use a dataset size of 500k.

| | Method | Action Err. Ratio ↓ | FVD ↓ | FID ↓ | SSIM ↑ | LPIPS ↓ | PSNR ↑ |
|---|---|---|---|---|---|---|---|
| Med. | AVID (Ours) (22M) | **1.355** | **57.4** | 10.77 | **0.592** | 0.270 | 19.9 |
| | Action-Conditioned Diffusion (22M) | 1.501 | 65.6 | 11.82 | 0.487 | 0.365 | 17.2 |
| | ControlNet-Small (22M) | 1.559 | 62.6 | **10.39** | **0.603** | **0.263** | **20.3** |
| Large | AVID (Ours) (71M) | **1.222** | **56.0** | 10.47 | 0.624 | 0.253 | 20.6 |
| | Action-Conditioned Diffusion (71M) | **1.224** | 60.0 | 11.16 | 0.558 | 0.312 | 18.5 |
| | ControlNet (71M) | 1.491 | 57.6 | **9.92** | **0.697** | **0.197** | **22.9** |
| Full | Pretrained Base Model (97M) | 3.855 | 204.0 | 13.03 | 0.451 | 0.352 | 17.3 |
| | Action-Conditioned Finetuning (97M) | **1.311** | **34.7** | **8.38** | **0.716** | **0.167** | **23.9** |

Table 4: Quantitative results for Coinrun 100k dataset. Shaded rows indicate that the method requires access to the model parameters.

| | Method | Action Err. Ratio ↓ | FVD ↓ | FID ↓ | SSIM ↑ | LPIPS ↓ | PSNR ↑ |
|---|---|---|---|---|---|---|---|
| Med. | AVID (Ours) (22M) | **1.277** | 24.7 | 8.01 | **0.679** | **0.177** | **23.0** |
| | Action-Conditioned Diffusion (22M) | 1.574 | 34.8 | 8.88 | 0.580 | 0.254 | 19.4 |
| | ControlNet-Small (22M) | 1.467 | **23.8** | **7.80** | 0.653 | 0.194 | 22.1 |
| Large | AVID (Ours) (71M) | **1.158** | **14.5** | **6.44** | 0.740 | 0.131 | 25.1 |
| | Action-Conditioned Diffusion (71M) | 1.203 | 17.7 | 6.63 | 0.704 | 0.158 | 23.4 |
| | ControlNet (71M) | 1.393 | 14.8 | 6.68 | **0.760** | **0.120** | **25.7** |
| Full | Pretrained Base Model (97M) | 3.855 | 204.0 | 13.03 | 0.451 | 0.352 | 17.3 |
| | Action-Conditioned Finetuning (97M) | **1.216** | **12.0** | **6.28** | **0.782** | **0.107** | **26.6** |

Table 5: Quantitative results for Coinrun 2.5M dataset. Shaded rows indicate that the method requires access to the model parameters.

## A.2 FULL RESULTS FOR AVID ABLATIONS

Table 6 contains ablation results for AVID across the full range of model sizes.

| | | Method | Action Err. Ratio ↓ | FVD ↓ | FID ↓ | SSIM ↑ | LPIPS ↓ | PSNR ↑ |
|---|---|---|---|---|---|---|---|---|
| Coinrun 500k | Med. | AVID (Ours) (22M) | **1.257** | **28.5** | 8.07 | 0.666 | 0.192 | 22.4 |
| | | No Mask (22M) | **1.241** | **28.6** | 8.06 | **0.675** | **0.186** | **22.7** |
| | | No Conditioning (22M) | 1.276 | 33.7 | 8.50 | 0.655 | 0.208 | 21.5 |
| | Large | AVID (Ours) (71M) | **1.154** | 23.1 | 7.33 | 0.713 | 0.161 | 23.8 |
| | | No Mask (71M) | **1.136** | **22.2** | **7.15** | **0.721** | **0.158** | **24.0** |
| | | No Conditioning (71M) | **1.141** | 27.8 | 8.02 | 0.682 | 0.189 | 22.3 |
| RT1 | Small | AVID (Ours) (11M) | **2.572** | **54.0** | 4.344 | **0.811** | **0.166** | **24.5** |
| | | No Mask (11M) | 2.752 | 63.7 | 4.566 | 0.809 | 0.168 | 23.8 |
| | | No Conditioning (11M) | 2.938 | 60.5 | 4.620 | 0.806 | 0.172 | **24.5** |
| RT1 | Med. | AVID (Ours) (34M) | **1.907** | **38.7** | 3.645 | **0.831** | **0.150** | 25.0 |
| | | No Mask (34M) | 2.155 | 49.7 | 3.940 | 0.825 | 0.156 | 24.5 |
| | | No Conditioning (34M) | 2.349 | 48.6 | 4.018 | 0.819 | 0.160 | **25.1** |
| RT1 | Large | AVID (Ours) (145M) | **1.609** | 39.3 | 3.436 | **0.842** | **0.142** | **25.3** |
| | | No Mask (145M) | 1.769 | 44.1 | 3.533 | 0.836 | 0.146 | **25.3** |
| | | No Conditioning (145M) | 1.775 | **36.2** | 3.550 | 0.827 | 0.149 | 25.0 |

Table 6: Full results for AVID ablations.

## A.3 RT1 RESULTS WITH LARGER COMPUTE LIMIT

The results in the main paper train models for RT1 using a compute limit of 7 days of 4× A100 GPUs. To provide reference values for performance we also evaluated IRASim (Zhu et al., 2024a) using our evaluation setup. IRASim is a 679M parameter model trained using 100 GPU-days on A800 GPUs. We also include results for action-conditioned finetuning of DynamiCrafter using a similar amount of compute (104 GPU-days on A100 GPUs). The results for these two models are in Table 7. By finetuning DynamiCrafter, we obtain slightly stronger performance than IRASim.

| Method | Action Err. Ratio ↓ | FVD ↓ | FID ↓ | SSIM ↑ | LPIPS ↓ | PSNR ↑ |
|---|---|---|---|---|---|---|
| IRASim | - | 21.0 | 2.759 | 0.879 | 0.1296 | 28.2 |
| Action-Conditioned Finetuning (1.4B) | 1.151 | 19.5 | 2.655 | 0.871 | 0.1160 | 27.1 |

Table 7: Quantitative results for RT1 dataset with greater computational budget.

## A.4 EXAMPLES OF AVID MASKING

Figures 5 and 6 illustrate examples of the mask generated by AVID which is used to mix the pre-trained model and adapter outputs.

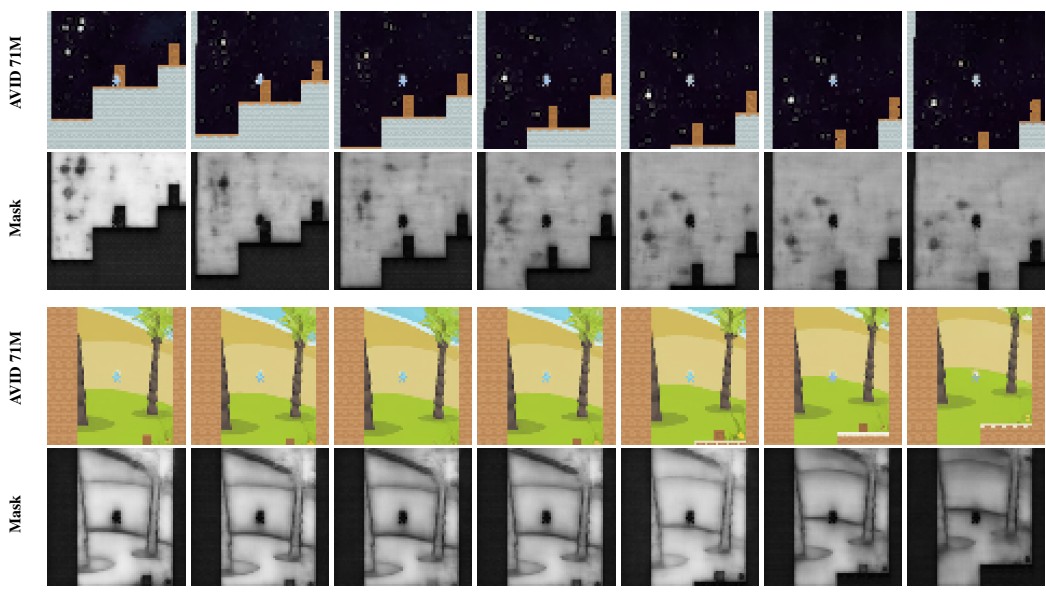

Figure 5: Examples of the mask, $m$, produced by AVID averaged throughout the diffusion process for Coinrun500k. White indicates the mask is set to 1, and black indicates the mask is set to 0.

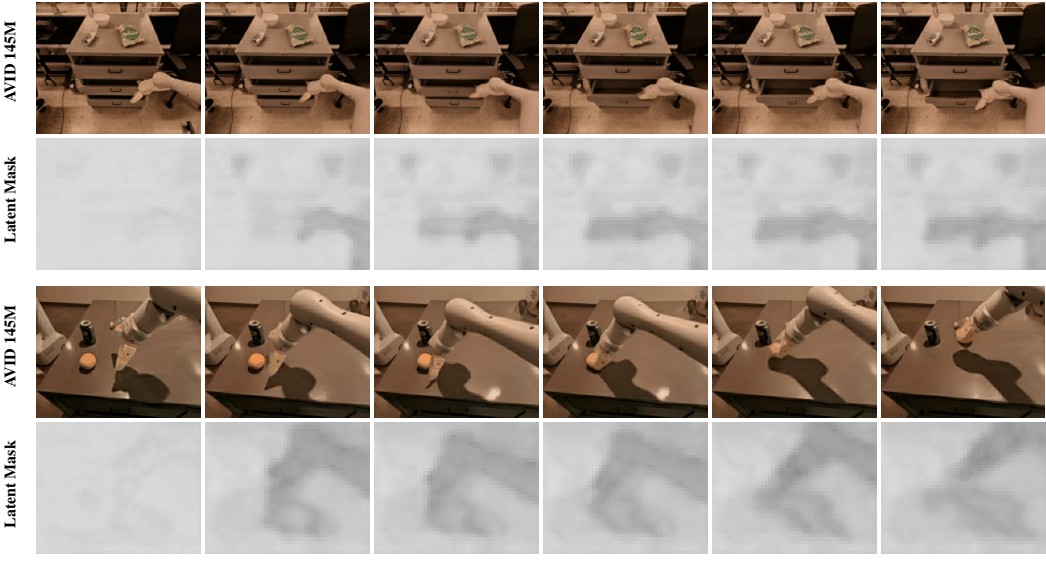

Figure 6: Examples of the mask, $m$, produced by AVID averaged throughout the diffusion process for RT1. Note that the mask is in the latent space, where the images have been downsampled by a factor of 8. White indicates the mask is set to 1, and black indicates the mask is set to 0.

## A.5 EXTENDED QUALITATIVE EXAMPLES

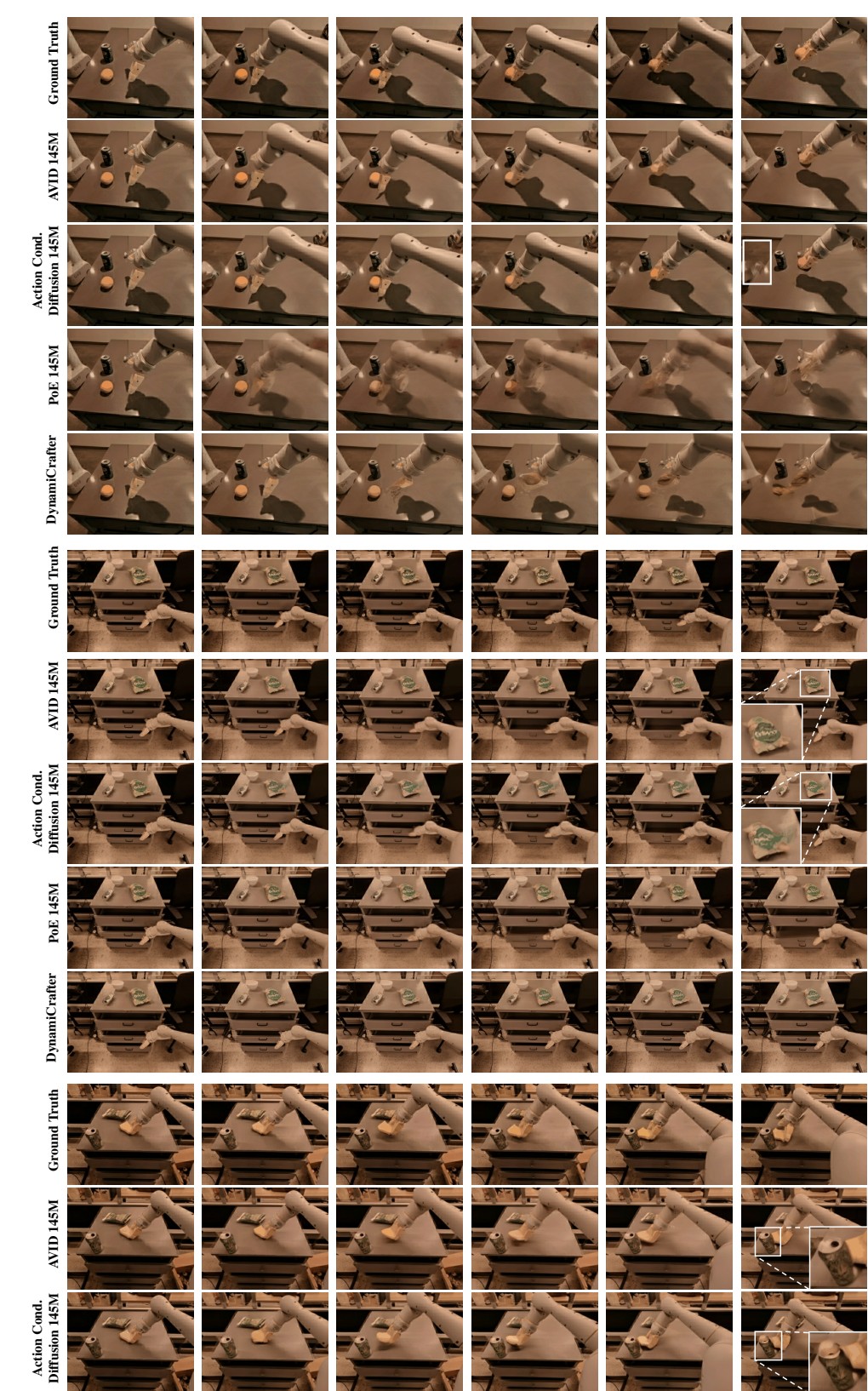

Figure 7: Extended qualitative comparison of videos generated for RT1.

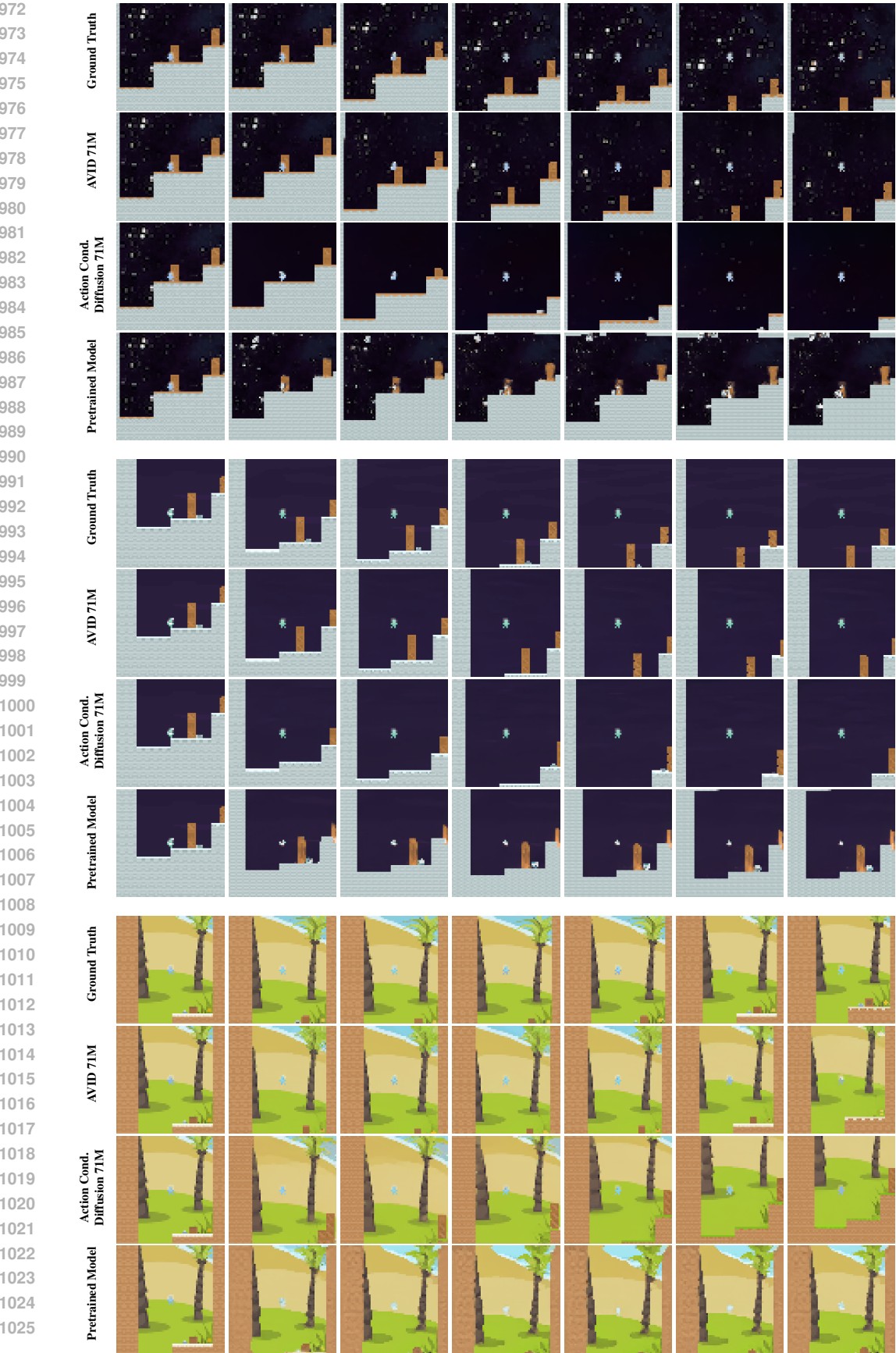

Figure 8: Extended qualitative comparison of videos generated for Coinrun 500k.

## A.6 EXAMPLE VIDEOS WITH SAME INITIAL FRAME

In Figure 9 we generate videos by providing the same initial conditioning frame but different actions. The provided action is fixed for all 16 steps of the video.

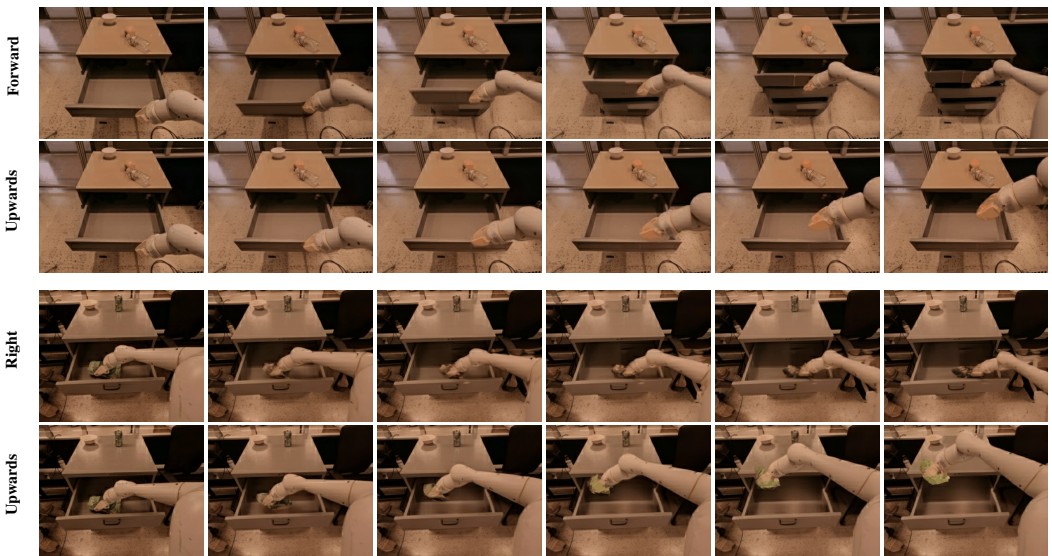

Figure 9: Examples of videos generated for RT1 with same initial frame but different actions.

## B EXPERIMENT DETAILS

### B.1 DATASET DETAILS

**Procgen Pretrained Model Dataset** The pretrained model is trained on videos sampled from 15 out of the 16 procedurally generated games of Procgen, excluding the 16th game Coinrun. Because the games are procedurally generated, each level in each Procgen game has different visual characteristics. To generate the dataset for the pretrained model, in each of the 15 games we sample an episode of 1000 steps long from each of the first 2000 levels by executing uniformly random actions. This results in 2M frames from each game for a total dataset of 30M steps, each with a resolution of $64 \times 64$. For training, we sample windows of 10 steps from the episodes. The pretrained model is trained on videos from this dataset for 12 days in a single A100.

**Coinrun Datasets** To train the Coinrun adapters, we generate an action-labelled dataset from the Coinrun game. At each timestep, the action is one of 15 discrete actions corresponding to keypad inputs. We sample episodes of 1000 steps from each of the first 100, 500, or 2500 levels by executing uniform random actions to create the Coinrun100k, Coinrun500k, and Coinrun2.5M datasets. For training, we sample action-labelled trajectories of 10 steps.

For evaluation, we create a held-out evaluation set of ground truth trajectories of videos and actions. We use 1024 ground truth trajectories of 10 steps each, sampled by executing random action sequences in levels sampled uniformly at random between level 10000 and 11000 of Coinrun. We use the same evaluation trajectories for all methods.

**RT1 Dataset** The RT1 dataset contains 87212 action-labelled episodes of a robot performing tasks such as picking up and placing objects, with a total of 3.78M steps. The action at each step is a 7 dimensional continuous vector corresponding to the movement and rotation of the end effector and opening or closing of the gripper. The videos have a resolution of $320 \times 512$. For training, we use 95% of the episodes, equating to 82851 episodes in the training dataset. We sample windows of 16 steps from these episodes for training the models. For evaluation, we use 1024 trajectories of 16

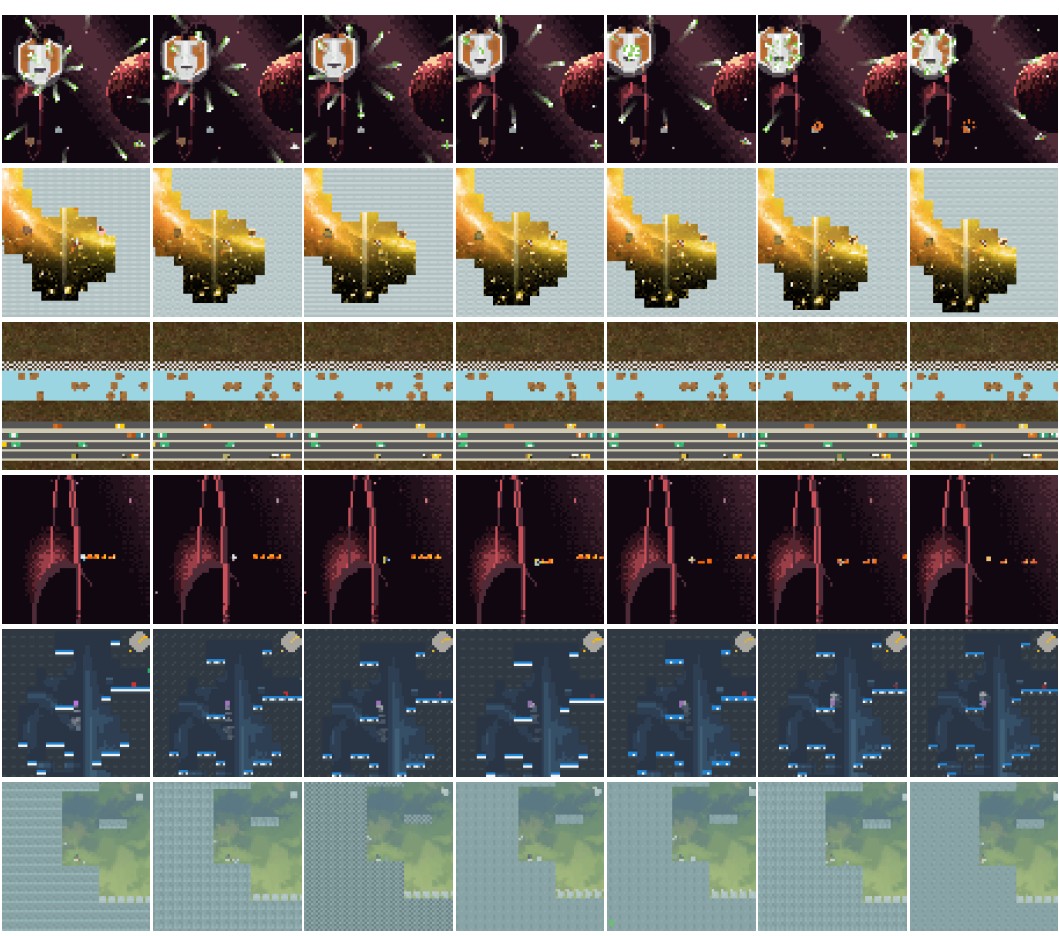

Figure 10: Examples from Procgen pretraining dataset on the 15 out of 16 Procgen games. Note that the pretraining dataset does not include any samples from Coinrun.

steps each sampled uniformly at random from the held-out test set of 4361 test episodes. We use the same evaluation trajectories for all methods.

## B.2 MODEL AND TRAINING DETAILS

**Procgen and Coinrun**    For both the pretrained model trained on 15 of the 16 Procgen games, and the adapters trained on the Coinrun datasets, we use the 3D UNet architecture from Video Diffusion Models (Ho et al., 2022c) trained on videos of resolution $64 \times 64$. We condition each model on two initial images to allow the initial direction the agent is moving in to be inferred and allow for more accurate video generation. Detailed hyperparameters for each of the models are in Table 8.

Table 8: Hyperparameters for models trained on Procgen and Coinrun.

| Hyperparameter | Value | Models |
|---|---|---|
| Base Channels | 100 | 97M Pretrained Model, ControlNet |
| | 90 | 71M |
| | 50 | 22M, ControlNet-Small 22M |
| Attention Head Dims | 64 | 97M Pretrained Model, all ControlNet variants |
| | 50 | 22M, 71M |
| Attention Heads | 8 | 97M Pretrained Model, all ControlNet variants |
| | 6 | 71M |
| | 4 | 22M |
| Learning Rate | 2e-5 | Finetuning |
| | 1e-4 | All other models |
| Training Time | 12 days | 97M Pretrained Model |
| | 3 days | All other models |
| Training Hardware | $1\times$ A100 GPU | |
| EMA | 0.995 | |
| Channel Multipliers | [1, 2, 4, 8] | |
| Sequence Length | 10 steps | |
| Batch Size | 64 | All |
| Noise Steps | 200 | |
| Inference Steps | 200 | |
| Sampling Method | DDPM | |
| Prediction Target | $\mathbf{x}_0$ | |
| Noise Schedule | Sigmoid | |

**RT1**    The pretrained model is DynamiCrafter (Xing et al., 2023), a latent video diffusion model. DynamiCrafter is trained to generate videos at a resolution of $320 \times 512$. To accommodate this, we resize and pad the images from the RT1 dataset to this resolution. DynamiCrafter is trained to optionally accept language conditoning. We use an empty language prompt, except in the case where the model is finetuned with language conditioning (see Section 4.1). DynamiCrafter uses the 1.4B 3D UNet architecture from (Chen et al., 2023a) and the autoencoder from Stable Diffusion (Rombach et al., 2022).

As discussed in Section 3.3, we use the same autoencoder for our AVID adapter, as well as all baselines. For all methods on RT1, we train a 3D UNet with the same architecture as Dynamicrafter from (Chen et al., 2023a), but with a reduced number of parameters. The hyperparameters for each of the models can be found in Table 9.

**Parameterisation**    We parameterise all models on Procgen and Coinrun to predict the clean video, $\mathbf{x}_0$, meaning that in practice the output of the pretrained model and the adapter output are both predictions of $\mathbf{x}_0$ rather than the noise. DynamiCrafter is trained using the $\mathbf{v}$ prediction target parameterisation (Salimans & Ho, 2022), so it outputs a prediction of $\mathbf{v}$. We also train all models on the RT1 dataset to predict the $\mathbf{v}$-target.

## B.3 ACTION ERROR RATIO EVALUATION METRIC DETAILS

**Coinrun**    The actions in Coinrun are discrete so to evaluate the Action Error Ratio we first train a classifier to predict the actions. The video is first processed using the encoder part of the 3D Unet

Table 9: Hyperparameters for models trained on RT1.

| Hyperparameter | Value | |
| --- | --- | --- |
| Base Channels | 320 | 1.4B, ControlNet |
| | 160 | ControlNet-Small 170M |
| | 96 | 145M |
| | 64 | 34M, ControlNet-Small 38M |
| | 32 | 11M, ControlNet-Small 10M |
| Attention Head Dims | 16 | 11M, ControlNet-Small 10M |
| | 64 | All other models |
| Channel Multipliers | [1, 2, 4, 4] | 1.4B, 145M, all ControlNet variants |
| | [1, 1, 2, 3] | 34M |
| | [1, 2, 3] | 11M |
| Learning Rate | 2e-5 | Finetuning |
| | 1e-4 | All other models |
| Attention Heads | Channels / Attention Head Dims | |
| Training Time | 7 days | |
| Training Hardware | $4\times$ A100 GPU | |
| EMA | 0.9995 | |
| Sequence Length | 16 steps | All |
| Batch Size | 64 | |
| Noise Steps | 1000 | |
| Inference Steps | 50 | |
| Sampling Method | DDIM | |
| Prediction Target | $\mathbf{v}$ | |
| Noise Schedule | Linear | |

architecture that we use for the diffusion model. This encoding is then flattened, and processed by an MLP which outputs a softmax over the 15 possible actions at each step of the video. The classifier is trained using the cross-entropy loss and has 60M total parameters. It is trained on a large dataset of 10M steps: 1000 steps sampled from each of 10,000 levels of Coinrun.

The Action Error Ratio is then computed as the ratio between the accuracy of the action classifier on real videos divided by the accuracy of the classifier on generated videos:

$$\texttt{ActionErrorRatio(discrete)} = \frac{\texttt{action\_accuracy(real\_videos)}}{\texttt{action\_accuracy(generated\_videos)}}$$

The action classifier achieves an accuracy of 0.267 on real videos from a held-out test-set. The accuracy score is low because not all of the 15 actions result in different outcomes in all states in Coinrun. Therefore it is not possible to predict actions at near 100% accuracy as the action taken is often ambiguous.

**RT1** The actions in RT1 are continuous so we train a regression model to predict the actions. The actions are first normalised to mean zero and unit variance. The video is processed by the encoder part of the 3D UNet architecture and then an MLP which outputs a prediction for the action at each time step. The regression model is trained using the MSE loss on the same 82851 video dataset that we use for training the adapter models, and has 85M parameters.

The Action Error Ratio is then computed as the ratio between the mean absolute error of the action predictions of the regression model on the generated videos compared to the real videos:

$$\texttt{ActionErrorRatio(continuous)} = \frac{\texttt{action\_MAE(generated\_videos)}}{\texttt{action\_MAE(real\_video)}}$$

The action error predictor achieves an MSE of 0.110 on a held-out test set of normalized actions.

## B.4 Normalized Evaluation Metric Details

To compute normalized evaluation metrics plotted in Figures 4a, 4b and 4c, we first normalize each evaluation metric to between 0 and 1. In RT1, 0 represents the worst performance for each metric across the Small, Medium or Large model sizes for Action-Conditioned Diffusion, ControlNet-Small, or AVID. 1 represents the best performance for each metric across these models. In Coinrun, 0 and 1 represent the worst and best performance for each metric across the Medium or Large model sizes for Action-Conditioned Diffusion, ControlNet/ControlNet-Small, or AVID, across all of the three datasets: Coinrun100k, Coinrun500k, and Coinrun2.5M.

The minimum and maximum values used for normalization are summarised in Tables 10 and 11.

|  | Action Err. Ratio | FVD | FID | SSIM | LPIPS | PSNR |
|---|---|---|---|---|---|---|
| **Minimum** | 1.384 | 24.9 | 3.375 | 0.767 | 0.142 | 22.9 |
| **Maximum** | 2.640 | 80.4 | 5.329 | 0.842 | 0.226 | 25.3 |

Table 10: Minimum and maximum values for metric normalization in RT1.

|  | Action Err. Ratio | FVD | FID | SSIM | LPIPS | PSNR |
|---|---|---|---|---|---|---|
| **Minimum** | 1.154 | 14.5 | 6.44 | 0.487 | 0.120 | 17.2 |
| **Maximum** | 1.574 | 65.6 | 11.82 | 0.760 | 0.365 | 25.7 |

Table 11: Minimum and maximum values for metric normalization in Coinrun.

If the goal is to maximize the metric, the normalized value of the metric is computed using:

$$\texttt{normalized\_value} = (\texttt{value} - \texttt{min\_value})/(\texttt{max\_value} - \texttt{min\_value}) \qquad (7)$$

If the goal is to minimize the metric, the normalized value of the metric is computed using:

$$\texttt{normalized\_value} = 1 - (\texttt{value} - \texttt{min\_value})/(\texttt{max\_value} - \texttt{min\_value}) \qquad (8)$$

Once the metrics have been normalized, we compute the mean across all 6 normalized metrics. This final value is plotted in Figures 4a, 4b and 4c.

## B.5 Baseline Details

Here we provide additional details about the baselines that are not included in the main paper.

- *Language-Conditioned Finetuning* – the language description that we use is: "Robot arm performs the task {task_description}" where {task_description} is the description given in the RT1 dataset.
- *Classifier Guidance* (Dhariwal & Nichol, 2021) – For the RT1 domain, since the actions are continuous, we discretise each action dimension into 256 bins uniformly following Padalkar et al. (2023) to train the classifier. We tune the weight of the guidance within $w \in \{0.003, 0.01, 0.03, 0.1, 0.3\}$.
- *Product of Experts* – we tune the weighting of the two models within $\lambda_p \in \{0.2, 0.4, 0.6\}$.
- *Action Classifier-Free Guidance* – we tune the weighting within $\lambda_a \in \{0.3, 1.0, 3.0, 10.0\}$.

For the baselines that require hyperparameter tuning, we sweep over each value of the hyperparameter and choose the value that obtains the best FVD in our evaluation.

