# OpenReview forum: "AVID: Adapting Video Diffusion Models to World Models"
_ICLR.cc/2025/Conference — Submitted to ICLR 2025_

### Official Review · Reviewer_Q4ad · 2024-10-29

**Soundness:** 3
**Presentation:** 4
**Contribution:** 3
**Rating:** 6
**Confidence:** 3

**Summary:**

This work proposes a method for leveraging diffusion models pretrained on large-scale action-free video data for training action-conditioned diffusion world models on smaller action-labeled data from domains of interest. The motivation for training these world models is to solve downstream sequential decision-making tasks. The proposed method’s main novelty is that it requires access to some intermediate calculations of the pretrained diffusion model but not to its parameters. The proposed method, AVID, trains an adapter network which is conditioned on the pretrained model’s noise prediction and optimizes a denoising loss that incorporates both noise predictions from the pretrained model and the adapter’s output using a learned mask. The author’s evaluate world model performance on a real-robot and a video game domain based on multiple perceptual metrics as well as an action-prediction-based metric. Baselines include various diffusion-based methods some of which require full access to model parameters. The proposed method either outperforms or is comparable to baselines in most of the evaluated metrics while not requiring access to pretrained model parameters.

**Strengths:**

Summarized points:
- Tackle an interesting problem in the path for scaling robot learning
- Clear and well written paper
- Good positioning in related work
- Comparison to relevant baselines
- Thorough analysis of results
- Detailed appendix

**Weaknesses:**

Summarized points:
- Intermediate model calculations are not necessarily more likely to be accessible than the model parameters
- Limitation analysis of previous work (Section 3.2) does not clearly motivate the author’s specific choice of solution
- Main motivation is sequential decision-making but evaluation metrics do not assess the world models’ efficacy in solving such tasks
- It is not clear from the experimental results that training from scratch is not preferable to the proposed method for downstream sequential decision-making

**Evaluation - Metrics**

The main motivation of your method is to accommodate sequential decision-making but evaluation metrics do not assess the world models’ efficacy in policy learning or planning.
All metrics excluding ‘Action Error Ratio’ are perceptual metrics that may be dominated by aspects of the videos that are not important for control. For this reason, I believe the most interesting and relevant metric out of the ones you display in your evaluation is the ‘Action Error Ratio’. Your evaluation could benefit from including additional metrics that are a better proxy for the world model’s usefulness in sequential decision-making. In the Procgen dataset for example, you may want to measure the ability to predict the reward from the generated frames as well as the actions.

I understand that evaluating the world models by actually using them to solve a sequential decision-making task may not be straightforward. Doing this for the RT1 dataset would be hard for multiple reasons, but it may be more feasible for the Procgen environments. One possible evaluation pipeline is training a separate model to predict the reward from a given frame and then use the cross-entropy method (CEM) or a similar sampling-based planning algorithm with model predictive control (MPC) on top of the world model to maximize the sum of rewards in the prediction horizon. Any decision-making algorithm you choose doesn’t have to be SOTA to demonstrate the point that a given world model is better than the other for this purpose.

What is the accuracy of the action predictor on each dataset? I believe this is important in order to validate the use of the ‘Action Error Ratio’ metric and that this information should at least be in the appendix.

**Evaluation - Baselines**

Why do you tune baseline hyperparameters based on FVD and not based on e.g. normalized evaluation metrics? I find this choice puzzling since you explicitly write in the results section that this metric is less suitable than others in the setting of action-conditioned video generation.

How do you choose which baselines out of the 8 you suggested appear in the result tables?

Can the authors please explain what is the purpose of the ‘Full’ row in the result tables?

**Evaluation - Results**

It is not clear from the experimental results that training from scratch is not preferable to the proposed method for downstream sequential decision-making, a point that is also suggested in the limitations section and is mostly based on the ‘Action Error Ratio’ metric. This is not to say that it clearly is not beneficial. I suggest adding a discussion about the differences in performance in the two domains which would incorporate further insights as to when and why training an adapter is preferable to training from scratch.

**Evaluation - Ablation Study**

*Mask ablation*: It is not clear from your results that the learned mask has performance benefits and can’t be ‘absorbed’ into the adapter noise prediction, especially since it hurts performance on one dataset and doesn’t in the other.
How do you explain the difference in the effects of the mask on performance in each dataset? I think a discussion with respect to factors like the relationship between pre-training and fine-tuning data in each dataset and with respect to the results presented in Figure 4d could shed more light on this matter.

*Conditioning ablation*: I think the method and/or ablation section can benefit from an explanation or intuition behind why conditioning on the pretrained model’s output is beneficial, given that the pretrained output is already accounted for in the objective.

*Request for ablation*: As I see it, the fundamental difference between the proposed method and the PoE baseline is that the parameters of the adapter network are trained on the denoising loss containing noise predictions from both the pretrained network and the adapter network. Therefore an interesting ablation would be combining both the NM and NC ablations.

**Questions:**

Most questions and suggestions are detailed in the 'Weaknesses' section.

**Limitations of Naive Adaptation of Yang et al.**

Can the authors please highlight the exact source of discrepancy between the derivation in Yang et al. to the derivation presented in this section? Do you claim that there is an error in their derivation? Alternatively, are there different assumptions in your setting where their derivation does not hold?

---

> ### Author Response · Authors · 2024-11-24
> **Response**
>
> Thank you for the valuable time you have spent reviewing our paper.
>
> **Evaluation - Metrics**
>
> Thank you for the suggestion to evaluate the world models for sequential decision-making using MPC. We agree that the experiment you proposed makes sense and would be very insightful. However, given the short time frame of the rebuttal period we have had to defer this to future work.
>
> We have added information about the accuracy of the action predictors on each dataset to appendix B3:
>
> Coinrun:
>
> “The action classifier achieves an accuracy of 0.267 on real videos from a held-out test-set. The accuracy score is low because not all of the 15 actions result in different outcomes in all states in Coinrun. Therefore it is not possible to predict actions at near 100\% accuracy as the action taken is often ambiguous.”
>
> RT1:
>
> “The action error predictor achieves an MSE of 0.110 on a held-out test set of normalized actions.”
>
> **Evaluation - Baselines**
>
> We agree that tuning the baseline hyperparameters based on the normalized evaluation metrics is a good idea. We originally used FVD as the tuning metric as it is a highly common video metric. Unfortunately, due to a change in institution the authors have been unable to obtain access to the raw results and so cannot reprocess the results using a different tuning criterion.
>
> We include all of the 8 baselines we suggested in Table 2 for the RT1 results. For the Table 1 Coinrun results, we do not include “Language-Conditioned Finetuning” as language conditioning is not relevant for that domain. Therefore, there are 7 baselines included in the Table 1 results.
>
> The “Full” row indicates models that do not fit within our Small/Medium/Large model groupings because we have not scaled down the trainable parameter counts to reduce model capacity. This includes:
>
> - The pretrained model
> - Full-finetuning methods
> - ControlNet for RT1 since this is much larger than the small/medium/large models.
> - Classifier guidance - for the simpler task of classifying actions, the classifier has a large number of trainable parameters and we therefore do not expect it to be limited by model capacity.
>
> **Evaluation - Ablation Study**
>
> Mask ablation: We cannot say for sure why the mask helps on one domain but not the other. However, a hypothesis is that the movements in Coinrun are discrete and the images are low resolution. Therefore, it is easy for the adapter model to correct incorrect motions output by the base model with or without the mask. In RT1, the images are higher resolution and the movements are more subtle and continuous. Therefore, the mask may make it easier for the model to learn the more challenging task of correcting the motion from the pretrained model in this domain.
>
> Conditioning ablation: We have added the following discussion to the ablations section. “Output conditioning enables AVID to observe errors in the pretrained output and then immediately make corrections. In contrast, NC has slower feedback to correct the pretrained model output as discussed in Zavadski et al. (2023).”
>
> Zavadski, Denis, Johann-Friedrich Feiden, and Carsten Rother. "Controlnet-xs: Designing an efficient and effective architecture for controlling text-to-image diffusion models." *arXiv preprint arXiv:2312.06573* (2023).
>
> Combining NM and NC ablations: Thank you for this suggestion. Unfortunately, due to a change in institution the authors no longer have access to compute so we are unable to run this experiment.
>
> **Limitations of Yang et al.**
>
> Since our submission, we were made aware of another paper which discusses this issue (Du et al. 2023). Du et al. 2023 also point out that the composition is incorrect. However, as t→0 the product model can be approximated by the sum of scores with error→0. Thus, Langevin dynamics with infinite steps and an annealed step size yields the correct distribution. However, the distribution is incorrect for the practical finite-step generation used in diffusion models.
>
> We have added this reference to the discussion in Section 3.2.
>
> Du, Yilun, et al. "Reduce, reuse, recycle: Compositional generation with energy-based diffusion models and MCMC." *International conference on machine learning*. PMLR, 2023.

---

> > ### Comment · Reviewer_Q4ad · 2024-11-25
> >
> > I thank the authors for their reply and for addressing some of my questions and concerns.
> >
> > In my opinion, it is a problem that the authors have been unable to obtain access to the results/resources they need to properly address the reviewers’ requests and concerns during the rebuttal period. It is very common for authors to run additional experiments and evaluation during this period and this limitation should have been resolved long before the actual rebuttal, assuming there are no extreme circumstances that prevented it.
> >
> > My remaining concerns are:
> >
> > - Downstream decision-making is not properly evaluated.
> > - It remains unclear if the author’s proposed method is preferable to training from scratch, nor is there a valuable discussion as to when and why this is the case.
> > - Baseline hyperparameter tuning based on FVD raises questions about the relevance of the results.
> > - The fact that there is ambiguity in action prediction given video frames in Coinrun, which results in a very low accuracy for the action classifier, weakens the use of the Action Error Ratio metric on this dataset. Given that AVID only performs better than training from scratch on this metric in Coinrun, this weakens the overall claim that AVID is indeed better than training from scratch.
> > - An ablation with NM and NC could provide more insight to the benefits of these components to your method.

---

### Official Review · Reviewer_Fgus · 2024-11-01

**Soundness:** 2
**Presentation:** 2
**Contribution:** 3
**Rating:** 6
**Confidence:** 3

**Summary:**

1. The authors propose a novel method to condition pre-trained video diffusion models on action sequences without access to the pre-trained model's parameters.
2. The authors demonstrate that their adaptation method is superior to the method proposed in "Probabilistic Adaptation of Text-to-Video Models" and mathematically highlight the limitations of this other approach.
3. The authors use different pre-trained base models and two different video domains, games and robotics to quantitatively evaluate their proposed method against the above adaptation approach and some other proposed baselines.

**Strengths:**

1. The authors propose a novel method to condition pre-trained video diffusion models on action sequences without access to the pre-trained model's parameters.
2. The authors mathematically highlight the limitations of the adaptation method proposed in "Probabilistic Adaptation of Text-to-Video Models" and  this other approach.
3. The authors demonstrate that their adaptation method has better action consistency compared to the other approach, using a new metric that they introduce.
4. The authors also propose multiple baselines to compare against their proposed method.

**Weaknesses:**

1. In Table 2, Action conditioned diffusion has a better Action Error Ratio compared to the proposed approach for all three (small, medium, large) variants. While the authors do note this as a limitation, this needs to be explained/investigated more. If it is better to just train an action conditioned diffusion model from scratch why should there be a need to adapt pre-trained models ?

2. Instead of using the action embedding to just scale and shift the t-th frame feature, have the authors explored using cross-attention layers directly with the action embedding sequence similar to language conditioning ? Are there any specific challenges that prohibit such an approach ?

3. It would be interesting to see results for each task type in RT-1 . Are there tasks that are much harder to model than others and what does that tell us about the approach ?

4. Some video visualisations of the generated videos (especially for robotics) would also be very useful to judge the effectiveness of the approach. Are the videos temporally consistent visually ?

5. Why is IRAsim's Action error ratio empty in Table 7 ? is it not possible to evaluate the Action Error Ratio of IRAsim ?

**Questions:**

see weaknesses.

---

> ### Author Response · Authors · 2024-11-24
> **Response**
>
> Thank you for the valuable time you have spent reviewing our paper.
>
> **Question 1**
>
> Our results show that adapting a pretrained model results in much more visually accurate video predictions. However, as you point out, the Action Error Ratio is slightly worse on the RT1 domain (although it is better on procgen). This is likely because the adapter has to correct inaccurate motions generated by the pretrained model, which in some domains may be more difficult than learning to generate the correct action-conditioned motion from scratch.
>
> In future work, we wish to explore which approach leads to the strongest performance on downstream decision-making tasks.
>
> **Question 2**
>
> We used the scale and shift conditioning as it has been shown to outperform cross-attention conditioning for diffusion transformers [1]. There is no reason that cross-attention couldn’t be used for the conditioning as an alternative, but we did not explore this.
>
> [1] Peebles, William, and Saining Xie. "Scalable diffusion models with transformers." *Proceedings of the IEEE/CVF International Conference on Computer Vision*. 2023.
>
> **Question 3**
>
> Thank you for this interesting suggestion. Unfortunately, due to changes in institution affiliations, the authors no longer have access to compute and model checkpoints so we are unable to regenerate the results with different task groupings.
>
> **Question 4**
>
> Video visualizations are provided in the powerpoint slides provided in the supplementary material. Please let us know if you would like us to provide the videos in an alternative format.
>
> **Question 5**
>
> We could not determine how the actions were normalized/preprocessed in the data provided by the IRASim authors. Therefore, we decided to omit the Action Error result, since the IRASim comparison is an auxiliary result that is not directly comparable to AVID.

---

> > ### Comment · Reviewer_Fgus · 2024-11-25
> >
> > I thank the authors for their reply and for addressing some of my concerns.
> >
> > Question 2
> >
> > I do not think that the DiT paper can serve as conclusive evidence that cross-attention based action conditioning in this paper's case would result in poorer performance. The DiT paper conditions on time step and class label not a more fine grained signal as a sequence of actions. My suggestion for trying a cross-attention based conditioning method is inspired by text-to-image/video diffusion methods where a sequence of tokens are used to condition the result in a fine-grained manner using cross-attention.
> > Unfortunately it seems the authors will not be able to test this out given their issues with compute.
> >
> > Question 3
> > It is unfortunate that the authors are unable to run new experiments during the rebuttal.
> >
> >
> > Given the circumstances, I will maintain my previous rating of this paper.

---

### Official Review · Reviewer_CD2R · 2024-11-03

**Soundness:** 3
**Presentation:** 2
**Contribution:** 2
**Rating:** 6
**Confidence:** 3

**Summary:**

This paper focuses on the problem setting of action-conditioned video generation. It proposes to adapt pre-trained video generation model to action labeled dataset without access to parameters from pre-trained models. The goal is to add action information as conditioning to pre-trained models for more accurate video predictions. Authors also analyze limitations in previous related work, production of expert, under a specific case. The proposed approach, AVID, trains an adapter in smaller size on action labeled video datasets. It takes noise predictions outputs from pretrained models and action information as input, and learns to output a mask that is used to combine outputs from pre-trained model and adapter. The adapter is trained with reconstruction loss between final output from both models and ground truth on domain-specific datasets. Authors conducted experiments on two robotics datasets, Procgen and RT1, and compared proposed approach to several baselines that have full access and do not assume access  to pretrained model parameters. Experiments results demonstrate that AVID outperforms baselines in generating more realistic videos and better quality given action information on these domains.

**Strengths:**

1. The paper is well written and easy to follow
2. The main idea of training a lightweight adapter for action-labeled domains is reasonable. It balances finetuning efficiency and task performance.
3. Baseline comparisons are comprehensive. Authors compared to many alternative baselines to demonstrate effectiveness of their approach. Authors provide qualitative visualizations for quality of generated videos and usefulness of learned masks.

**Weaknesses:**

1. The presentation in Section 3.2 is a little unclear. It is hard to connect analysis about limitations of previous work [1] to motivations of the proposed approach
2. The novelty is somewhat limited. The main difference from previous work is to have domain-specific adapter output an element-wise mask that is used to combine noise predictions from pre-trained model and adapter.
3. The experimental domains are only two datasets within action-conditioned world modeling

[1] Yang, Mengjiao, et al. "Probabilistic adaptation of text-to-video models." arXiv preprint arXiv:2306.01872 (2023).

**Questions:**

1. Regarding W1, can authors elaborate more on how and why this analysis motivates design choices in methodology of AVID?
2. In the ablation study of “No mask” (Table 3), is the adapter trained with $\epsilon_{\text{final}}$ given in Equation 5 or the adapter not being able to output a mask?
3. Since the mask is an important component in design choices of AVID, could author visualize what the mask looks like in different timesteps of diffusion denoising process, which corresponds to Figure 4d?
4. Since AVID performs two diffusion denoising process, does this increase inference time and thus limit the scope of downstream applications of synthetic videos generated from this approach?
5. Regarding W3, is it technically possible to apply this approach to domains other than world modeling?

---

> ### Author Response · Authors · 2024-11-24
> **Response**
>
> Thank you for the valuable time you have spent reviewing our paper.
>
> **Question 1**
>
> [1] argues for their proposed PoE method because it enables a diffusion model to be adapted without access to the weights of the denoising model. However, per Section 3.2 we show that this does not correctly optimize for the denoising objective.
>
> Our approach resolves this issue by making the simple insight that under the same assumptions (access to pretrained model outputs only), given the pretrained model we can train an adapter to directly optimize the denoising objective (via Equation 6), thus overcoming the issue of composing independently trained models in Section 3.2.
>
> **Question 2**
>
> For “No Mask” the adapter is not able to output a mask during both training and during inference.
>
> **Question 3**
>
> Unfortunately, due to changes in institution affiliations, authors no longer have access to compute or model checkpoints so are unable to create this visualization. We agree that this would be an interesting visualization, and regret that we are unable to do this.
>
> **Question 4**
>
> Yes, the inference time is increased to approximately 1.5x the original pretrained model (it is less than 2x because the adapter model is smaller). We have added a short comment on this to the limitations section.
>
> **Question 5**
>
> Yes, there is nothing specific about approach that makes it applicable only to action-conditioning for world modelling. Our approach could in general be used to add a new conditioning signal to a pretrained model. We have added a comment on this for future work in the conclusion: “We also wish to explore using AVID adapters to add new conditioning signals to pretrained models other than actions.”

---

> > ### Comment · Reviewer_CD2R · 2024-11-26
> >
> > I would like to thank authors for answering my questions. This addresses most of my concerns, especially providing more detailed discussions and comparisons to PoE [1]. My remaining suggestions to this work are extending to different downstream tasks other than action conditioning in world models and improving efficiency in inference time due to the use of additional adapter. Changed to weak accept.

---

> ### Comment · Area_Chair_hmwt · 2024-11-26
> **[ACTION NEEDED] Respond to author rebuttal**
>
> Dear Reviewer,
>
> Now that the authors have posted their rebuttal, please take a moment and check whether your concerns were addressed. At your earliest convenience, please post a response and update your review, at a minimum acknowledging that you have read your rebuttal.
>
> Thank you,
> --Your AC

---

### Official Review · Reviewer_NTHp · 2024-11-03

**Soundness:** 2
**Presentation:** 3
**Contribution:** 2
**Rating:** 5
**Confidence:** 4

**Summary:**

The paper proposes a mechanism for adapting current image-conditioned video diffusion models to action-conditioned video diffusion models. They do this by training an additional UNet after the standard video diffusion UNet which predicts an adjustment to the noise output by the standard UNet. Because of this setup, their "adapter" does not need access to the parameters of the pretrained video diffusion model. Experiments show that this kind of noise adaptation helps for some metrics and does not for some others.

**Strengths:**

- The paper has a nice motivation: how does one adapt existing foundation models in order to add an action conditioning to them, so as to make it more relevant and useful for embodied robotics applications
- The paper writing is clear; first the limitations of prior work are built up and then a solution is proposed

**Weaknesses:**

- The way the paper starts with the motivation near L51-52 is a bit misleading. The paper actually cannot fix the issues in L51-52 because they still assume access to the internal inference pipeline of these closed-source model, because if I understand it correctly, this method needs access to a diffusion model's noise prediction at each of the N reverse diffusion steps that happens at inference. For closed source models, this information is not available.
- The performance gain in the quantitative metrics is not substantial. The metrics where the proposed method shines are mostly photometric quantities. It is not clear if the error margin between prior work and this work just results from the standard deviation or variance of the models. I think a better reflection of the proposed approach would have come from an application to a downstream robot task (maybe manipulation) that would evaluate a robot in action. PSNR and other photometric errors with the shown gain do not say much about the performance of the method.
- The paper heavily leans on the limitations of the POE approach and that is how a learnable adapter is motivated but qualitatively there is no comparison to that approach (even though POE is slightly better than the action conditioned diffusion across some metrics and settings).

**Questions:**

See weaknesses.

---

> ### Author Response · Authors · 2024-11-24
> **Response**
>
> Thank you for the valuable time you have spent reviewing our paper.
>
> **Motivation in Introduction**
>
> You are correct that our approach requires access to intermediate model outputs which are not available in current closed source models. We have modified the introduction in lines 70-71 to make it clear that our approach is not directly applicable to current closed source model: “We advocate for providers of closed-source video models to provide access to intermediate model outputs in their APIs to facilitate the use of adaptation approaches such as AVID.”
>
> **Performance for Downstream Tasks**
>
> The improvement in framewise-prediction errors demonstrates that AVID results in more accurate video generation, and we would expect that this translates to more accurate decision-making for downstream tasks. We aim to apply AVID to a real decision-making task in future work.
>
> **Qualitative Comparison to PoE**
>
> We have added a qualitative comparison to PoE in Figure 7 in appendix A.5. The videos from PoE are much more blurry, and one of them looks like two different videos superimposed. We have added the following comment to the qualitative results section (Line 320):
>
> “The videos generated by PoE are blurry, and sometimes appear like two superimposed videos.”

---

> > ### Comment · Reviewer_NTHp · 2024-11-25
> >
> > Thank you for adding a qualitative comparison. Unfortunately I'm not convinced that AVID would show higher performance improvement for downstream applications. If there was evidence for AVID being suitable for adding any other modality of conditioning to video diffusion models, that would have been another reason to accept the paper. And given that there is also no analysis on standard deviation of experiments, I would like to maintain my previous rating.

---

### Author Response · Authors · 2024-11-24
**Thank you**

We thank the reviewers for their insightful reviews, and have updated our paper accordingly. We will respond to each review separately.

---

### Meta-Review · Area_Chair_hmwt · 2024-12-20

**Metareview:**

This is a well-written paper about training an action-conditioning adapter for a pre-trained, frozen video diffusion model.

The reviewer reception of this paper was mixed: while the writing and presentation are clear and of high quality, there were concerns about the generality of the method and the extent of the experimental evaluation to fully justify the claims made in the paper. For example, it would have been valuable to demonstrate the proposed adapter not just for action conditioning, but for other forms of controlling video diffusion (or even image diffusion) models.

The AC agrees with these concerns and recommends refining the evaluation of the method should the authors wish to submit a revised version of the paper to a future venue.

**Additional Comments On Reviewer Discussion:**

No reviewer was willing to champion the paper for acceptance.

---

### Decision · Program_Chairs · 2025-01-22

Reject